# EXSEQREG: Explaining sequence-based NLP tasks with regions with a case study using morphological features for named entity recognition

Onur Güngör *, Tunga Güngör, Suzan Uskudarli

Computer Engineering Department, Boğaziçi University, Istanbul, Turkey

* onurgu@boun.edu.tr

## Abstract

The state-of-the-art systems for most natural language engineering tasks employ machine learning methods. Despite the improved performances of these systems, there is a lack of established methods for assessing the quality of their predictions. This work introduces a method for explaining the predictions of any sequence-based natural language processing (NLP) task implemented with any model, neural or non-neural. Our method named EXSEQREG introduces the concept of region that links the prediction and features that are potentially important for the model. A region is a list of positions in the input sentence associated with a single prediction. Many NLP tasks are compatible with the proposed explanation method as regions can be formed according to the nature of the task. The method models the prediction probability differences that are induced by careful removal of features used by the model. The output of the method is a list of importance values. Each value signifies the impact of the corresponding feature on the prediction. The proposed method is demonstrated with a neural network based named entity recognition (NER) tagger using Turkish and Finnish datasets. A qualitative analysis of the explanations is presented. The results are validated with a procedure based on the mutual information score of each feature. We show that this method produces reasonable explanations and may be used for i) assessing the degree of the contribution of features regarding a specific prediction of the model, ii) exploring the features that played a significant role for a trained model when analyzed across the corpus.

## Introduction

In recent years, machine learning methods have been successful in achieving the state-of-the-art results in many natural language processing tasks (NLP), mainly due to the introduction of neural models. As such, numerous novel architectures have been proposed for virtually every task. Although the ability to account for biases or explain the predictions is just as important as the accuracy, clear and satisfying explanations for the success are often not addressed.

**Data Availability Statement:** The following data were used for training the models employed during the use case and the subsequent evaluation of the method. The joint Turkish NER and MD dataset

used in this study is available at the following repository: https://github.com/onurgu/joint-ner-and-md-tagger/tree/xnlp/dataset. The joint Finnish NER and MD dataset is available at the following repository: https://github.com/onurgu/joint-ner-and-md-tagger/tree/xnlp/dataset.

**Funding:** This research was partially supported by Bogazici University Research Fund (BAP) to TG and SU (Grant 13083). Sahibinden.com provided financial support in the form of a contract-based salary for OG. The specific roles of this author are articulated in the 'author contributions' section. The funders had no role in study design, data collection and analysis, decision to publish, or preparation of the manuscript. No additional external funding was received for this study.

**Competing interests:** The authors have read the journal's policy and the authors of this manuscript have the following competing interests: OG is a paid employee of Sahibinden.com. The numerical calculations reported in this paper were partially performed at TUBITAK ULAKBIM, High Performance and Grid Computing Center (TRUBA resources). There are no patents, products in development or marketed products associated with this research to declare. This does not alter our adherence to PLOS ONE policies on sharing data and materials.

Various approaches to provide explanations for machine learning predictions have been proposed [1–5]. One of the promising approaches to explain the outcome of a machine learning model is Local Interpretable Model-Agnostic Explanations (LIME) [6], which attempts to explain a model's prediction based on the model's features. The given input sample is perturbed by randomly removing some features. Consequently, the model's prediction function is employed to obtain probabilities corresponding to the perturbed versions of the input sample. LIME is based on the idea that the prediction probabilities of these perturbed samples can be modeled by a linear model of features. The solution of this linear model gives a vector of real values corresponding to the importance of each feature. Such vectors are considered to be valuable in assessing the quality of a model since they render the insignificant features evident, which are often the culprits in biased decisions.

In this work, we propose an extended version of LIME to handle any sequence-based NLP task in which a procedure for transforming the task into a multi-class classification problem can be constructed. This method utilizes *region*s which refer to the segments of the inputs that are directly related to the predictions, e.g. the tokens that cover a named entity in named entity recognition. The transformation procedure requires the probability of each prediction associated with a given region. Some models yield these probabilities as a part of their output. In other cases, access to the internals of the model are required to compute these probabilities. For example, for the sentiment classification task, typically a vector of class potentials is used to predict the sentiment of a sentence. This vector is employed to calculate the probability of each sentiment for the given sentence. For tasks with more complex labels, further computation may be required to calculate the probability of each prediction. For example, the probability of the entity tag for named entity recognition task is computed with the probabilities of the token-level tags. In the extension we propose for LIME, perturbations are performed by removing each feature of a region independently as opposed to selecting several features randomly. The prediction probabilities of each label in these perturbed samples are calculated using the transformation procedure specific to the task as detailed in the Perturbation and Calculating probabilities sections.

The main aim of the proposed method is to provide a vector whose values indicate the strength and the direction of the impact of each feature. The first step is to observe the probability differences caused by the removal of each feature due to the perturbations. A linear regression model is used to relate these differences with the removed features. The solution of this linear model gives a list of weights corresponding to each feature which can be regarded as the impact of each feature on the current prediction, which we consider to be an explanation.

We demonstrate the method on named entity recognition (NER) task by using a NER tagger trained on Turkish and Finnish both of which are morphologically rich languages [7, 8]. The tagger requires all the morphological analyses of each token in the sentence to be provided to the model. Fig 1 shows a sentence in Turkish with the potential morphological analyses for the named entity "Ali Sami Yen Stadyumu'nda" (means 'at the Ali Sami Yen Stadium' in English) that covers the tokens from $2^{nd}$ to $5^{th}$ position. The model predicts the correct morphological analyses, which it utilizes to recognize the named entities.

In the Analysis section, we provide quantitative and qualitative evaluations of the results of the explanation method. The quantitative evaluation compares the most influential morphological features in predictions with those whose mutual information scores are the highest with respect to entity tags. The qualitative assessment involves an analysis of the morphological tags relevant to several named entity tags for Turkish and Finnish.

Our contributions can be summarized as:

<div style="text-align:center">

**Henüz** **Ali** **Sami** **Yen** **Stadyumu'nda** **oynamamıştı.**

**1** **2** **3** **4** **5** **6**

LOCATION

</div>

**2** ali+Noun+Prop+A3sg+Pnon+Nom

**3** sami+Noun+Prop+A3sg+Pnon+Nom

**4** ye+Verb^DB+Verb+Pass+Pos+Imp+A2sg
yen+Noun+Prop+A3sg+Pnon+Nom
yen+Noun+A3sg+Pnon+Nom
yen+Verb+Pos+Imp+A2sg

**5** stadyum+Noun+Prop+A3sg+P3sg+Loc
stadyum+Noun+Prop+A3sg+P2sg+Loc

**Fig 1. A Turkish sentence (translated as "She/he had not played at the Ali Sami Yen Stadium yet.") with one named entity tag and the possible morphological analyses of the tokens in the named entity.**

1. a general method to explain predictions of any sequence-based NLP task by means of transforming them into multi-class classification problems,

2. a method to assess the impact of perturbations of input samples that relies on probability differences instead of the typical use of exact probabilities,

3. an encoding that distinguishes whether a feature absent in the perturbed sample was present in the original input, thereby capturing the knowledge of a removal operation,

4. a qualitative and quantitative evaluation of the proposed method for the NER task for the morphologically rich languages Turkish and Finnish, and

5. an open source software resource to replicate all results reported in this work [9].

The remainder of this paper is organized as follows: the Background section provides information required to follow the proposed method. After we relate our work to the current literature in the Related work section, we give the details of the method in the Explaining sequence-based NLP tasks section. The CAnalysis section details the results of applying our method to Turkish and Finnish NER datasets. Finally, the Conclusions and Future Directions section summarizes the main takeaways and contributions of this work along with future directions.

## Background

### LIME

Local Interpretable Model-Agnostic Explanations (LIME) [6] is a method for explaining the predictions of any machine learning model and is agnostic to the implementation of the model. It treats the model as a blackbox that produces a prediction along with an estimated probability. LIME belongs to the class of methods called additive feature attribution methods [10]. These methods yield a list of value pairs composed of a feature and its impact on the prediction. This list is regarded as an explanation of the prediction based on the magnitude and

the direction of the impact. Typically, these methods learn a linear model of the features to predict the expected probability of the prediction. The data samples required to train the linear model are obtained by perturbing the original input sample by randomly removing a feature.

In order to represent the features that are removed or retained during the perturbation, a binary vector $z$ that is mapped to the original input $x$ with a function $h$ is used. The mapping depends on the model to be explained. For example, if a model expects the input sentence $x$ to be in the bag-of-words form, $x$ consists of word and frequency pairs. In this case, the binary vector $z$ is composed of $z_i$s each of which indicate whether or not the $i^{\text{th}}$ word is retained. In other words, if $z_i$ is 1, word $i$'s bag of words frequency value remains the same as the frequency in the input sentence, otherwise it is set to zero.

Additive feature attribution methods are generally defined as

$$g(z) = \phi_0 + \sum_{i=1} \phi_i z_i \tag{1}$$

where $z_i$ is the binary value that indicates whether feature $i$ is retained or not, $\phi_i$ is a value that indicates the importance of feature $i$, and $\phi_0$ is the bias. The function $g(z)$ is the outcome of the linear model that estimates the probability $f(x)$ which is obtained from the machine learning model. The following function is minimized to obtain the importance values $\phi_i$:

$$\underset{g}{\text{argmin}}\ L(f, g, \Pi(x, z)) + \Omega(g)$$

where $f$ is the probability function of the model, $\Pi(x, z)$ is the local weighting function and $\Omega(g)$ constrains the complexity of $g$. For example, to explain a text classification model, one might set $\Pi(x, z)$ to an exponential kernel with cosine distance between $x$ and $z$. Any function that satisfies the distance constraints, namely the non-negativity, zero distance if $x = z$, symmetry, and the triangle inequality ($d(x, z) \leq d(x, y) + d(y, z)$), may be used for $\Pi(x, z)$. A reasonable choice for $\Omega(g)$ is a function that returns the number of words in the vocabulary. Accordingly, the loss function $L$ is defined as the sum of the squared errors weighted by $\Pi(x, z)$:

$$L(f, g, \Pi(x, z)) = \sum_{x, z=h(x)} \Pi(x, z)(f(x) - g(z))^2.$$

## A neural NER tagger model

To demonstrate the method proposed in the Analysis Section, we use a tagger model [7] that jointly addresses the named entity recognition (NER) and the morphological disambiguation (MD) tasks. Fig 2 shows this model while processing a sentence fragment. Each word is represented as a fixed-size vector which is fed to the first layer of the sentence-level Bi-LSTM (bidirectional long short-term memory). A fixed-size vector representation for each possible morphological analysis of each word $i$ is computed by a separate Bi-LSTM layer (not shown in the figure). These vectors (depicted as rectangles with a surrounding dashed line) which are denoted by $\text{ma}_{ij}$ are multiplied with the context vector $h_i^1$ which is the output of the first layer of the sentence-level Bi-LSTM component. The model selects the one that has the maximum multiplication result, $\text{ma}_{ij^*}$ which becomes the disambiguated morphological analysis of the $i^{\text{th}}$ word.

Each level $l$ of the sentence-level Bi-LSTM is fed with the concatenation of the previous level's output $h_i^{l-1}$ and the original word representation. The output of the final layer of the sentence-level Bi-LSTM component $\bar{h}_i^3$ is concatenated with the most probable morphological

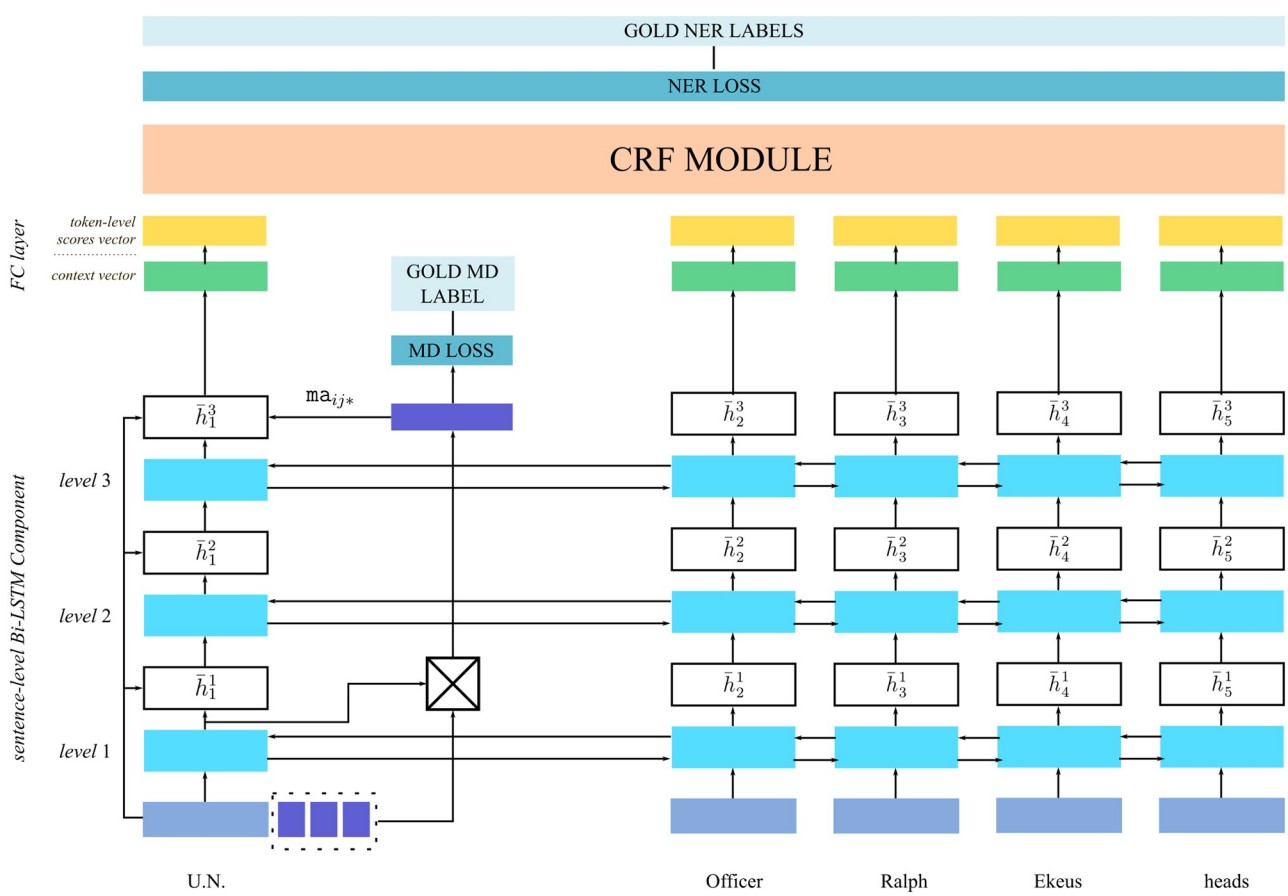

**Fig 2. Joint neural NER and MD model processing the sentence fragment "U.N. Officer Ralph Ekeus heads"** [7].

analysis' vector $\mathrm{ma}_{ij*}$ and fed into a fully connected (FC) layer to obtain the score vectors for each word. These score vectors denote the model's estimated score for each token-level entity tag to be the correct one for that position. These are then employed by a conditional random field (CRF) layer to decode the most probable path among all possible paths of token-level entity tags. Finally, the sequence of token-level entity tags form the output of the NER tagger for the whole sentence.

## IOBES tagging scheme for named entity recognition

Named entities are labeled with types, such as 'PER' for person and 'LOC' for location. The IOB (Inside-Outside-Beginning) scheme uses particular prefixes for each token within a chunk to indicate whether the token is inside (I), outside (O), or beginning (B) of the named entity [11]. The IOBES scheme extends the IOB scheme to indicate the ending token and the single tokens with the 'E' and 'S' prefixes, respectively.

The labels are formed of the position prefix followed by '-' and the type of the entity. Thus, a named entity of 'LOC' type consisting of a single token would be labeled as 'S-LOC'. A named entity of 'LOC' type that consists of three tokens would be labeled as 'B-LOC', 'I-LOC', and 'E-LOC' in this order. All tokens that are not a part of any named entity are labeled as 'O'.

## Morphologically rich languages

In morphologically rich languages, the morphology of words express a significant amount of grammatical information as opposed to other languages. This is realized by affixing the root words with morphemes that convey syntactic information. For example, possession is indicated using a suffix in Turkish (such as 'araba (car) + -ım', yielding 'arabam' meaning 'my car'), whereas the same meaning is conveyed by the use of a word in English. Morphologically rich languages utilize affixes frequently to produce valid word forms, which renders morphological analysis very significant for such languages.

Various notations have been introduced to analyze the structure of derived words in morphologically rich languages. In our case, we utilize the notation introduced by Oflazer [12] for Turkish and the Universal Dependencies Project [13] for Finnish.

## Related work

There are several approaches to explaining the results of machine learning models. Some machine learning models are self-explanatory, such as decision trees, rule-based systems, and linear models. For example, the output of a decision tree model is a sequence of answers to yes/no questions, which can be considered as explanations. For other models, special mechanisms should be designed to provide explanations. Explanation models aim to provide two types of explanations: i) model (or global) explanations, and ii) outcome (or local) explanations. Local explanations focus on the outcome resulting from specific input samples, whereas model explanations reveal information about the machine learning model in question. Explanation models further differ in their explanation methods, the types of machine learning models that can be explained, and the type of data that can be explained [14]. According to the classification in [14], the method proposed in this work is a model-agnostic *features importance* explainer, since it aims to reveal the importance of each feature given an input sample.

The method proposed in this work (EXSEQREG) is inspired by the LIME approach [6] which explains the predictions of any model. It achieves this by perturbing the input to assess how the predictions change. LIME uses a binary vector to indicate whether a feature is perturbed, as described in the LIME section. The binary vector ($z$) indicates the presence or absence of a feature. One shortcoming of this representation is that it does not convey whether a feature that is absent in the perturbed version is due to removal, since it may have been absent in the original input. In other words, a zero value may indicate two states: the feature does not exist at all or it is perturbed. Since this distinction may be significant in an explanation, we modified this scheme to remedy this. We follow an encoding scheme where we mark the features that are present but removed with minus one, the features that are present and not removed with one, and the features that are not present at all with zero. Furthermore, we focus on the probability differences induced by perturbations as opposed to the exact probabilities that are utilized by LIME.

A recent approach called LORE [15] learns a decision tree using a local neighborhood of the input sample. The method then utilizes this decision tree to build an explanation of the outcome by providing a decision rule to explain the reasons for the decision and a set of counterfactual rules to provide insights about the impacts of the changes in the features.

The method proposed in our work is part of a general class of methods called additive feature attribution methods [10]. Methods from this family include DeepLIFT [16], layer-wise relevance propagation method [2], LIME [6], and methods based on classic Shapley value estimation [1, 4, 17]. DeepLIFT aims to model the impact of altering the values assigned to specific input parts. Layer-wise relevance propagation works similar to DeepLIFT, however, in this case, the altered values are always set to zero. Shapley value estimation depends on the average

of prediction differences when the model is trained repeatedly using training sets perturbed by removing a single feature $i$ from a subset $S$ of all unique features. Sampling methods for efficiently computing Shapley values are also offered [4]. All these methods, including LIME, depend on solving linear models of binary variables similar to Eq 1.

An explanation method for NER based on LIME has been proposed by [18]. The method treats input sentences as word sequences and ignores fine-grained features such as part-of-speech tags which are often attached to words as part of the input. The resulting explanation is a vector of real numbers that indicate the impact of each word. The method is restricted to models that are limited to token-level named entity tag prediction. However, every token is dependent on each other in the named entity recognition task. Many models exploit this dependency and combine token-level named entity tag prediction probabilities to have a single named entity prediction probability for the entire token sequence of the entity. Contrary to this method, in this work, we aim to handle this dependency issue by proposing a special transformation procedure, which is detailed in the Calculating probabilities section for named entity recognition.

LIME defines a text classification problem conditioned on features that correspond to the frequency of unique words in the input sentence. To obtain the explanation vector for a given prediction, the input is perturbed by selecting a random set of words and eliminating all instances from the input sentence, thus removing the bag-of-words frequency values for the corresponding words from the input. This causes problems while explaining models that employ sequence processing constructs like RNN because a bag-of-words feature is devoid of information about the positions of the words within a sentence. This makes it difficult to relate a specific feature to a certain portion of the input sentence. Instead, selecting random subsequences (or substrings if we ignore tokenization) from the input sentence, designating these as distinct features, and removing these new features would both perturb the original sentence and enable specifying the specific position of the perturbation. These position-aware features are used in another extension of LIME to explain the prediction of such models [19]. In this work, however, we are only interested in the impact of features in a specific region in the sentence, so it is not required to have position-aware features.

The interpretation of machine learning models for NLP became significant subsequent to the success achieved by neural model. Although they achieve state-of-the-art results for many tasks, their black box nature leaves scientist curious about whether these models learn relevant aspects. For non-neural models, the approach was to provide mechanisms for explanations of features and their importance. However, the complexity of neural models has rendered explanations for models or specific outcomes very difficult. One approach is to use auxiliary diagnostic models to assess the amount of linguistic knowledge that is contained in a given neural representation [20–24].

Another prominent approach is to exploit attention mechanisms in the models [25, 26] to explain specific outcomes by attaching importance values to certain input features, like n-grams, words, or characters that make up the surface forms. Most of these methods modify the input samples so that they are reflected as changes in the output or the inner variables of the models. Other works exploit specially created datasets to assess the performance of an NLP task. For example, a custom dataset derived from a corpus of tasks related to the theory of mind was used to explore the capacity of a question answering model to understand the first and second-order beliefs and reason about them [27]. Custom datasets are also used when trying to test whether the semantic properties are contained in word representations by using a special auxiliary diagnostic task that aims to predict whether the word embedding contains a semantic property or not [28].

Finally, approaches have been proposed to explain machine learning models by introducing latent variables to models [29] or that produce inherently interpretable output such as via word alignment information in machine translation [30].

## Explaining sequence-based NLP tasks

This section introduces a method for explaining specific predictions of models trained for sequence-based NLP tasks. Essentially, the method provides explanations about which part of the input impacts the prediction of a given neural model. The method produces an explanation vector of scores that indicate the impact of the features used by the model. This vector can be utilized in offering an explanation to the user of the model's prediction. For example, a model trained for classifying the sentiment of a sentence may rely on features such as the specific words that occur in the sentence, the position and the number of punctuation marks in the sentence, or the content of the fixed-size vector representations pretrained for each word. In this sentiment classification task, the user should be suspicious of a model if words that have clear negative sentiment are effective in a positive sentiment prediction.

## Defining NLP tasks

We define NLP tasks as *processes* that transform input consisting of a sequence of tokens along with a set of features into a sequence of labeled tokens. Fig 3 provides an overview of NLP processes. In this work, we denote the input with $X$, the tokens for input with $T$ and for output with $T'$, the number of tokens for input with $N_t$ and for output with $N_o$, the output labels with $Y$, and the number of output labels with $N_y$. These *processes* are implemented by *models*. Each *model* has a *prediction function* that maps $X$ to $T'$ and $Y$. The model architecture determines the size and the contents of the feature sets $F$. An example for a type of feature could be the word embedding that corresponds to a token in $T$. This is a flexible definition that applies to nearly all NLP tasks.

Some NLP tasks and their corresponding parameters are shown in Table 1. The sentiment classification task can be associated with the question: "Is the sentiment of the sentence X positive?". The expected output is simply "Yes" or "No". In this case, there are no token outputs, thus $N_o = 0$ and the cardinality of the label space is two (or $|Y| = 2$). The word sense disambiguation task can be expressed with the question "What is the sense of the word $X$?" whose answer

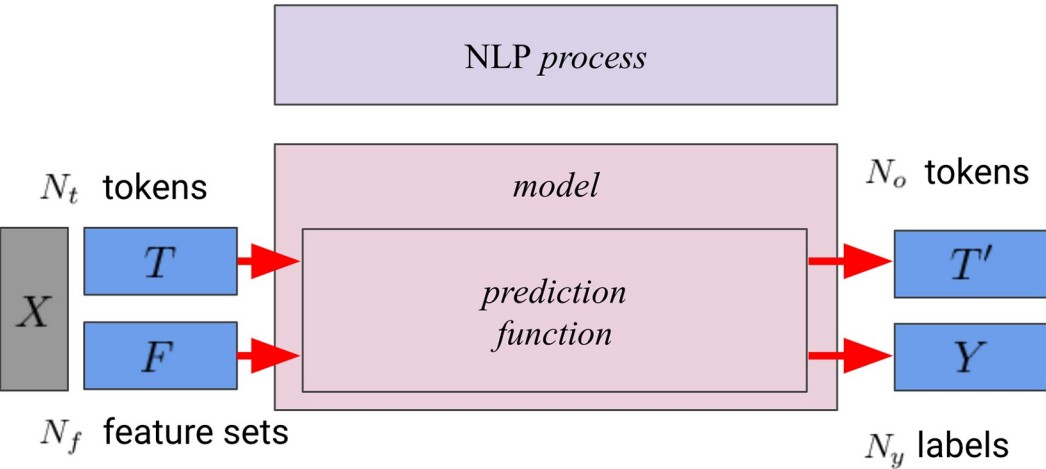

**Fig 3. Relations between processes, models, input, and output in this work.**

**Table 1. Selected examples of NLP tasks that can be covered by the method.**

| Task | $T$ | $F$ | $T'$ | $Y$ |
|---|---|---|---|---|
| Sentiment classification (model 1) | "Great music!" $N_t = 2$ | A feature set for each token, $N_f = 2$ | $N_o = 0$ | 'YES' $N_y = 1$ |
| Sentiment classification (model 2) | "Great music!" $N_t = 2$ | A feature set for the whole sentence, $N_f = 1$ | $N_o = 0$ | 'YES' $N_y = 1$ |
| Word sense disambiguation | "We bought gas for the **car**." $N_t = 6$ | A feature set for each token, $N_f = 6$ | $N_o = 0$ | *Sense* = automobile $N_y = 1$ |
| Named Entity Recognition (in Turkish) | "Henüz **Ali Sami Yen Stadyumu** taşınmamıştı.", $N_t = 6$ | A feature set for each token, $N_f = 6$ | $N_o = 0$ | "O B-LOC I-LOC I-LOC E-LOC O" $N_y = N_t = 6$ |
| Machine translation (from Turkish to English) | "Henüz Ali Sami Yen Stadyumu taşınmamıştı.", $N_t = 6$ | A feature set for each token, $N_f = 6$ | "Ali Sami Yen Stadium was not relocated yet." $N_o = 8$ | $N_y = 0$ |
| Morphological disambiguation (in Turkish) | "Henüz Ali Sami Yen Stadyumu taşınmamıştı.", $N_t = 6$ | A feature set for each token, $N_f = 6$ | "Henüz Ali Sami Yen Stadyum +u taşın+ma+mış+tı" $N_o = 6$ | "Henüz Ali Sami Yen Stadyum+Acc taşın+Neg+PastPart+Past", $N_y = 6$ |

is one of the expected word senses. In this case, again, $N_o = 0$ while $N_y = 1$, but the cardinality of the output label space is the number of possible senses. The NER task can be expressed as a mapping from each input token to a named entity tag. As such, $N_o = 0$, $N_y = N_t$ and the cardinality of the output label space is the number of token tag sequences of length $N_y$. Machine translation can also be defined by this scheme by setting $N_t > 0$, $N_o > 0$, $N_y = 0$, and the cardinality of the output token space to the number of token sequences of length $N_o$. In another case, the task may require both output tokens and output labels, like in morphological disambiguation which we give an example in Table 1. This scheme is flexible enough to express models that employ features concerning the whole sentence. For example, an alternative version of the example for sentiment classification could have a single feature set for the whole sentence (e.g. sentence embedding). The parameter configuration for this case would be $N_t > 0$, $N_f = 1$, $N_o = 0$, and $N_y = 1$.

## EXSEQREG: Explaining sequence-based NLP tasks with regions

This section describes the proposed framework for explaining neural NLP models. For illustration purposes, we use a specific NER tagger as a use case. We describe our method using a set of variables along with the indices $i$, $t$, $j$, and $k$ (see Table 2). These indices are used to refer to input sentence $X_i$ as the $i$th sentence in the dataset, feature set $F_{it}$ corresponding to $t$th token in sentence $i$, and label $Y_{it}$ corresponding to $t$th token in sentence $i$. Fig 4 depicts an example that utilizes these variables. For the NER model, $Y_{it}$ is the token-level named entity tag, e.g. 'B-PER', 'I-PER', 'I-LOC', and similar. The input to the NER tagger consists of the morphological analyses, the word embeddings and the surface forms of the tokens. The NER tagger

**Table 2. Summary of variables used by the explanation method.**

| Variable | Description |
|---|---|
| $i, X$ | input sequence index and input sequence |
| $t, T$ | token index and token sequence |
| $f, F$ | feature and set of features |
| $Y$ | label sequence |
| $j, R$ | region index in sentence and set of regions |
| $\pi, \Pi$ | perturbed sentence and set of perturbed sentences |
| $k, K$ | label $k$ and number of labels |
| $e_{ij}^{k\star}$ | explanation vector for region $j$ in sentence $i$ for label $k$ |

| $T_{it}$ | Amazon | Web | Servicesin | pääevankelista | Jeff | Barr | kertoo | |
|---|---|---|---|---|---|---|---|---|
| $t$ | 1 | 2 | 3 | 4 | 5 | 6 | 7 | $N_t = 7$ |
| $Y_{it}$ | B-PRO | I-PRO | E-PRO | O | B-PER | E-PER | O | $N_y = 7$ |

PRODUCT — Region $r_{i1}$     PERSON — Region $r_{i2}$

| $t$ | Morphological analyses | Feature sets |
|---|---|---|
| 1 | - Amazo\|~NOUN~N~Case=Gen\|Number=Sing <br> - Amaz\|~NOUN~N~Case=Ill\|Number=Sing <br> - Amazo\|~PROPN~N~Case=Gen\|Number=Sing <br> - Amazon\|~PROPN~N~Case=Nom\|Number=Sing <br> - Amazon\|~ADJ~A~Case=Nom\|Degree=Pos\|Number=Sing <br> - Amazko\|~NOUN~N~Case=Gen\|Number=Sing <br> - Amazon\|~PROPN~N~ <br> - Amazto\|~NOUN~N~Case=Gen\|Number=Sing <br> - amazon\|~NOUN~N~Case=Nom\|Number=Sing | $F_{i1}$ <br><br> Case=Gen <br> Number=Sing <br> Case=Ill <br> Case=Nom <br> Degree=Pos |
| 2 | - Web\|~PROPN~N~Case=Nom\|Number=Sing <br> - Web\|~PROPN~N~ <br> - Web\|~PROPN~N~Case=Nom\|Number=Sing <br> - Web\|~NOUN~N~Abbr=Yes\|Case=Nom\|Number=Sing <br> - web\|~NOUN~N~Case=Nom\|Number=Sing | $F_{i2}$ <br><br> Case=Nom <br> Number=Sing <br> Abbr=Yes <br> Case=Nom |
| 3 | - Services\|~PROPN~N~Case=Gen\|Number=Sing <br> - Servicesi\|~PROPN~N~Case=Gen\|Number=Sing <br> - Servicesin\|~ADV~Adv~ <br> - Servicess\|~PROPN~N~Case=Gen\|Number=Sing\|Typo=Yes <br> - Servicest\u00e4\|~VERB~V~Mood=Ind\|Number=Sing\|Person=1\|Tense=Past\|VerbForm=Fin\|Voice=Act <br> - Servicet\u00e4\|~VERB~V~Mood=Ind\|Number=Sing\|Person=1\|Tense=Past\|VerbForm=Fin\|Voice=Act <br> - Servicet\u00e4\u00e4\|~VERB~V~Mood=Ind\|Number=Sing\|Person=1\|Tense=Past\|VerbForm=Fin\|Voice=Act | $F_{i3}$ <br><br> Case=Gen <br> Number=Sing <br> Typo=Yes <br> Mood=Ind <br> Person=1 <br> Tense=Past <br> VerbForm=Fin <br> Voice=Act |

| $\mathbb{F}_{i1} = \cup_{t=1}^{3} F_{it}$ | $\mathbf{e}_{i1}^{\text{PRO}\star}$ | |
|---|---|---|
| Feature | − | + |
| Number=Sing | ▨ | |
| Case=Nom | | ▨ |
| Case=Gen | | ▪ |
| Tense=Past | | ▪ |
| Typo=Yes | | ▪ |
| Degree=Pos | | ▪ |
| Person=1 | | ▪ |
| VerbForm=Fin | | ▪ |
| Mood=Ind | | ▪ |
| Abbr=Yes | | ▪ |
| Case=Ill | | ▪ |
| Voice=Act | | ▪ |

**Fig 4. Explanation of a NER tagger's prediction where $N_t = 7$, $N_f = 7$, $N_y = 7$, $N_o = 0$. $X_i$ is the $i$th sentence.** Only first seven tokens of the sentence are shown due to space constraints which is translated as "Jeff Barr, the chief evangelist of Amazon Web Services, says . . .". Region $r_{i1}$ is labeled as a 'PRO' entity tag. All possible morphological analyses of 1, 2, and 3 tokens ($t$) in region $r_{i1}$ that are used to form feature sets $F_{it}$ are shown. The union of all $F_{it}$ in region $r_{i1}$ yields the set of morphological tags $\mathbb{F}_{i1}$. $\mathbf{e}_{i1}^{\text{PRO}\star}$ is the resulting explanation vector.

exploits the information conveyed by the morphological tags within the analyses. The details of the NER tagger can be found in the A neural NER tagger model section.

The method proposes the concept of region, which is used to refer to a specific part of the input sentence. For example, for the NER task, regions refer to named entities which may span several consecutive tokens. Regions are used to associate features and predictions related to a segment of the input. Fig 4 shows a Finnish sentence with two regions marking named entities, one of type 'PRODUCT' and the other of type 'PERSON'.

We define an explanation vector $\mathbf{e}_{ij}^{k\star}$ to explain the prediction of label $k$ for the $j^{\text{th}}$ region of sentence $i$. This vector's length equals the number of unique features in the model. The regions are denoted by a sequence of integers that give the positions of the tokens belonging to the region. For the NER problem, $k$ is the named entity tag and the number of unique features is equal to number of unique morphological tags in the model. The values of the dimensions of $\mathbf{e}_{ij}^{k\star}$ represent the impact of their corresponding features.

A full example of the NER task is depicted in Fig 4 where $T_i$ is the sequence of words in sentence $X_i$ and $N_t$ is the number of words. There are no output tokens for this task, thus $N_o = 0$. There is a label and a list of morphological analyses corresponding to each input token, making $N_y = N_t = N_f$. The regions which contain each named entity are denoted as $r_{ij}$. As shown in the figure, $r_{i1}$ spans the first three tokens, 'Amazon Web Servicesin', and is labeled with named entity tag 'PRO' which signifies a product name. This can be seen by observing the $Y_{it}$ values which are the token-level named entity tags. The lower part of the figure lists all possible morphological analyses for each token $t$ and its feature sets $F_{it}$ originating from these lists. The union of every $F_{it}$ in the region $r_{i1}$ is denoted as $\mathbb{F}_{i1}$. The explanation vector $\mathbf{e}_{i1}^{\text{PRO}\star}$ in the lower right part of the figure contains a real value for each feature in $\mathbb{F}_{i1}$. The diagram indicates that 'Case = Nom' contributes positively to the 'PRO' prediction. On the other hand, the presence of the 'Number = Sing' and 'Case = Gen' morphological tags is expected to decrease the probability of the 'PRO' named entity tag. This is a simple example where the explanations can be viewed as the degree to which a morphological tag is responsible for identifying a specific named entity tag.

The remainder of this section describes the main steps required to calculate $\mathbf{e}_{ij}^{k\star}$:

1. Perturbing $X_i$ to obtain a set of sentences $\Pi_i$ (the Perturbation section).

2. Calculating probability changes corresponding to all regions $r_{ij}$ of $X_i$ using sentences in $\Pi_i$ (the Calculating probabilities section).

3. Defining and solving a special regression problem corresponding to every region $r_{ij}$ in every perturbed sentence in $\Pi_i$ (the Computing importance values section).

## Perturbation

NLP tasks are divided into several classes according to their region types. The widest regions span entire sentences, such as in the case of sentiment classification. The regions within sentences may be contiguous or not. For example, the NER task is almost always concerned with contiguous regions but the co-reference resolution task or the multi-word expression detection task is usually characterized by noncontiguous regions.

In all of these cases, the $j^{\text{th}}$ region in sentence $i$ is denoted as $r_{ij}$ and represented by a sequence of integers that correspond to the positions of the tokens in the region. We define $R_i$ as the set of all regions $r_{ij}$ in $X_i$. For every $r_{ij}$ in $R_i$, we perturb $X_i$ by only modifying the features that are found in that region. A region $r_{ij}$ in the NER task is represented with a sequence of integers, i.e. ($start$, ..., $end$) where $start$ and $end$ are the first and last position indices of the words in the region. For example, in the Turkish sentence "Henüz Ali Sami Yen Stadyumu'nda oynamamıştı", there exists a single region which spans the words 2 to 5, i.e. (2, 3, 4, 5).

The set of features subject to perturbation in region $r_{ij}$ is defined as $\mathbb{F}_{ij} = \cup_{t \in r_{ij}} F_{it}$. We perturb $X_i$ by independently removing each feature $f \in \mathbb{F}_{ij}$ from $X_i$ to obtain $\pi_{ij} = \{\texttt{remove}(X_i, j, f) : f \in \mathbb{F}_{ij}\}$. The expression $\texttt{remove}(X_i, j, f)$ denotes a sentence originating from sentence $X_i$ where all instances of feature $f$ are removed from all $F_{it}$ in

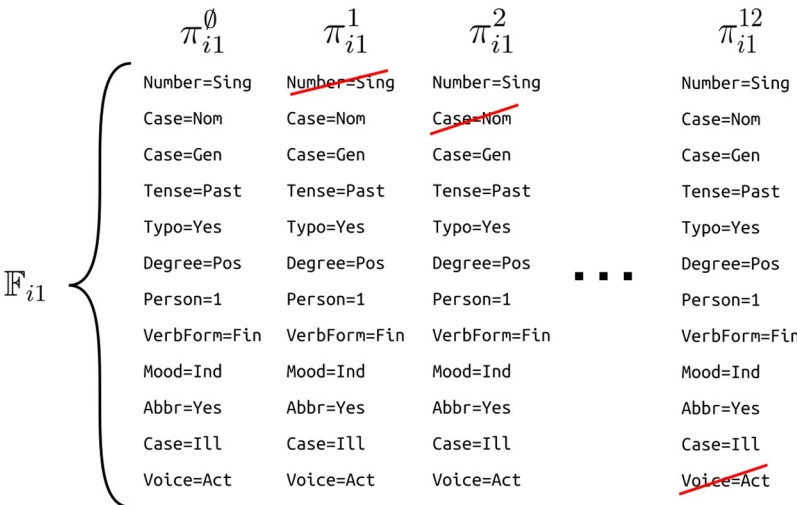

**Fig 5. The change in $\mathbb{F}_{i1}$ corresponding to every perturbed sentence $\pi_{i1}^r$ is depicted.** Crossed out morphological tags are removed from $\mathbb{F}_{i1}$. Morphological tags which are not already present are not shown.

region $r_{ij}$. The unperturbed version of $X_i$ is denoted as $\pi_{ij}^{\emptyset}$. Fig 5 shows the change in $\mathbb{F}_{ij}$ corresponding to each perturbed version $\pi_{ij}^r$. To form $\pi_{i1}^1$, the morphological tag 'Number = Sing' is removed from the morphological analyses of tokens 1, 2, and 3 (e.g. 'Amazo|∼ NOUN∼N∼Case = Gen|Number = Sing' to yield 'Amazo|∼NOUN∼N∼Case = Gen'). The collection of $\pi_{ij}$'s results in a set $\Pi_i$ consisting of at most $\sum_{j'=1}^{|R_i|} |\mathbb{F}_{ij'}|$ perturbed sentences, which is at most $|R_i| \times |M|$ where $M$ is the set of unique features in the model. Unlike the named entity recognition task where $M$ is low, this perturbation strategy might be problematic for other tasks where the number of unique features is very high. For example, the number of features that are constructed combinatorially from input segments become very large as sentence lengths increase. In such cases, the feature to be removed could be selected in a uniformly random manner from $\mathbb{F}$. This is repeated for several times to form a set of perturbed samples with a feasible size.

Eventually, a set of perturbed samples $\pi_{ij}$ for each region $r_{ij}$ is obtained to be used as input to the prediction function of the model.

## Calculating probabilities

In this step, we seek to obtain a matrix of label prediction probabilities $P_{ij}$ where the $r^{\text{th}}$ row corresponds to the $r^{\text{th}}$ perturbed version of $X_i$ in $\pi_{ij}$, namely $\pi_{ij}^r$. Each row of $P_{ij}$ is a vector $p_{ij}^r$ of length $K$ where each dimension corresponds to a label $k$ of the task at hand. Thus the size of $P_{ij}$ is $|\pi_{ij}| \times K$.

Depending on the task and the model, $p_{ij}^r$ might be computed by the model itself. For instance, a sentiment classification model might yield the probability of the positive label directly. On the other hand, it might be necessary to compute $p_{ij}^r$ using some output of the model. Some models include a component which indirectly corresponds to the prediction probability of each label $k$ in a region $r_{ij}$. For example, the model implementing the NER task (described in the A neural NER tagger model section) does not directly output the probability of the presence of a named entity in a given region. The model aims to find the contiguous sequence of tokens referring to named entities in the given sentence. To do this, it assigns a

score to each possible IOBES tag attached to each word in the sentence. It then selects the most probable sequence of tags over all possible sequences of IOBES tags for the sentence. However, for explanation purposes, we are only interested in the labels of the named entity regions $r_{ij}$.

To provide an explanation for the prediction in region $r_{ij}$ for tasks that are not classification problems, we need a mechanism for transforming them into multi-class classification problems. For the NER task, the IOBES tags in the region must be transformed to named entity tags. The transformation procedure selects paths satisfying the following regular expression "S-TAGTYPE | B-TAGTYPE,[I-TAGTYPE]*,E-TAGTYPE | O+". The resulting path list is filtered so that it only includes paths with a single entity. We omit paths that result in multiple entities or paths that are invalid (e.g. starting with a 'I-' prefix) in the region as the trained model consistently attaches very low probabilities to such cases. For other NLP tasks, one should start with enumerating all the possible prediction outcomes in a given region $r_{ij}$. If the number of total outcomes in a region is very high, it is advised to omit the outcomes which are expected to have very low probabilities. After this filtering, each remaining outcome is considered as a label.

In Fig 6, we present the correct sequence of tags for a Turkish named entity tag 'LOC'. This is one of the $13^4$ possible sequences. The total number of possible sequences is calculated by multiplying the number of possible token-level tags at each token position $t$. In this case, the total number of possible sequences is calculated as $(4^*K + 1)^N$ where $K$ is equal to the number of entity types and $N$ is the number of tokens.

After this transformation procedure, the NER task which is originally a sequence tagging problem is reduced to a classification problem with $K$ classes. The NER model involves score

**Fig 6. A NER task for a Turkish sentence (meaning "Ali Sami Yen Stadium had not been relocated yet.").** The token-level tag predictions for each position are shown with their scores $s_{t,o}$. The correct sequence of the token-level named entity tags are marked in red. Single token entities are not applicable to multiple token entities, thus their scores are N/A.

variables $s_{t,o}$ which are used to predict IOBES tags ($o$) for each position $t$, as depicted in Fig 6. During normal operation, the NER model feeds these scores to a CRF layer. The CRF layer treats these scores as token-level log-likelihoods and uses the learned transition likelihoods to choose the most probable path (see the A neural NER tagger model section). For the purposes of the explanation method, we define the probability of the sequence corresponding to entity tag $k$ in named entity region $r_{ij}$ as

$$p_{ij}^r(k) = P(k|\pi_{ij}^r) = \frac{\exp\left(\mathtt{score}(\pi_{ij}^r, k)\right)}{Z_{ij}^r}$$

where $\mathtt{score}(\pi_{ij}^r, k)$ is the total score of entity tag $k$ using the score variables ($s_{t,o}$) in the region and $Z_{ij}^r$ is $\sum_{k'} \exp\left(\mathtt{score}(\pi_{ij}^r, k')\right)$. We also define the same probability for region $r_{ij}$ in the unperturbed sentence $X_i$ and refer to it as $p_{ij}^\emptyset$.

## Computing importance values

The previous steps of the method produce a $\pi_{ij}$ and $P_{ij}$ for each region $r_{ij}$. The final step aims to produce an explanation for every label $k$ for every region $r_{ij}$. An explanation $e_{ij}^k$ for label $k$ of region $j$ in sentence $X_i$ is a vector which contains one dimension for each feature in the model. Each element of $e_{ij}^k$ indicates the impact of a feature $m$ from $M$ for predicting label $k$, where $M$ is the set of features used in the model. Note that a region in a given sentence $X_i$ is not always related to all features, thus the number of features $|\mathbb{F}_{ij}|$ related to region $r_{ij}$ is usually smaller than $|M|$. This guarantees $|M| - |\mathbb{F}_{ij}|$ dimensions to be zero. For example, for the NER task in this work, most of the morphological tags are not related to all regions.

We add the original sentence $X_i$ to $\pi_{ij}$ together with $|\mathbb{F}_{ij}|$ perturbed sentences to obtain a set of $|\mathbb{F}_{ij}| + 1$ sentences. We first form a matrix $\mathbb{C}_{ij}$ of size $(|\mathbb{F}_{ij}| + 1) \times |M|$ where the $r$th row corresponds to $\pi_{ij}^r$ if $r \leq |\mathbb{F}_{ij}|$. The last row of $\mathbb{C}_{ij}$ corresponds to the unperturbed version of $X_i$. Every row of $\mathbb{C}_{ij}$ is composed of ones, minus ones, and zeros signifying whether the feature that corresponds to the $m^{\text{th}}$ position was i) present and retained, ii) present and removed, and iii) was not present in input, respectively. We chose this scheme rather than using only ones and zeros to mark presence and absence because the latter one penalizes the features that were present but removed by perturbation.

Secondly, we form a matrix $\Delta\mathbb{P}_{ij}$ of size $K \times (|\mathbb{F}_{ij}| + 1)$. The last column is set to $\vec{0}$ as there is no perturbation in the original sentence, and the $r^{\text{th}}$ column of the first $|\mathbb{F}_{ij}|$ columns is equal to $p_{ij}^r - p_{ij}^\emptyset$. In other words, the entry $(k, r)$ of $\Delta\mathbb{P}_{ij}$ contains the difference induced in the probability of predicting label $k$ after the perturbation described by the row $r$ of $\mathbb{C}_{ij}$.

The matrices $\mathbb{C}_{ij}$ and $\Delta\mathbb{P}_{ij}$ are then combined in the loss function of ridge regression which employs regularization on the explanation vector

$$||\Delta\mathbb{P}_{ij}(k, :) - \mathbb{C}_{ij} e_{ij}^k||_2^2 + ||e_{ij}^k||_2^2 \qquad (2)$$

and minimized with respect to $e_{ij}^k$. We use notation $A(i, :)$ to refer to the $i^{\text{th}}$ row of matrix $A$. We call the corresponding solution as $\mathbf{e}_{ij}^{k\star}$. We give the pseudo-code of the method in Algorithm 1.

Let us consider a very simple task, and assume that there are two features *Non-emotional* and *Emotional*. There are three labels *Positive, Negative*, and *Neutral*. Thus $|M|$ is 2 and $K$ is 3. Let's further assume that the sentence $i$ consists of a single region which spans the whole sentence and both features are present in this region. This gives us a single region $r_{i1}$ and as there are two features $|\mathbb{F}_{i1}|$ is 2. Thus $\Delta\mathbb{P}_{i1}$ is of size $3 \times (2 + 1)$ and $\mathbb{C}_{i1}$ is of size $(2 + 1) \times 2$. Let's

choose the entries of these matrices so that we observe that the probability of predicting '*Positive*' label for the sentence increases when

- feature *Non- emotional* was present and removed, and

- *Emotional* was present but not removed.

This is represented in the first row of $\mathbb{C}_{i1}$ (i.e. [−1, 1]) and in $\Delta\mathbb{P}_{i1}(1, 1)$ (i.e. 0.3) in the following equations. We chose the other values so that the probability of predicting '*Positive*' decreased (i.e. −0.1) when feature '*Emotional*' was present and removed, and feature '*Non-emotional*' was present but not removed.

$$\Delta\mathbb{P}_{i1}(1,:) \quad = \begin{bmatrix} 0.3 & -0.1 & 0 \end{bmatrix}^{\mathrm{T}} \tag{3a}$$

$$\mathbb{C}_{i1} \quad = \begin{bmatrix} -1 & 1 \\ 1 & -1 \\ 1 & 1 \end{bmatrix} \tag{3b}$$

According to the definitions above, we can define the explanation vector for label *Positive* as

$$\mathbf{e}_{i1}^{1\star} = \underset{e_{i1}^{1}}{\operatorname{argmin}} ||\Delta\mathbb{P}_{i1}(1,:) - \mathbb{C}_{i1} e_{i1}^{1}||_{2}^{2} + ||e_{i1}^{1}||_{2}^{2}. \tag{4}$$

When we solve it, we obtain

$$\mathbf{e}_{i1}^{1\star} = \begin{bmatrix} -0.08 & 0.08 \end{bmatrix}^{T} \tag{5}$$

which can be interpreted as feature '*Non-emotional*' has a negative impact on the prediction of label '*Positive*', but '*Emotional*' has an opposite impact.

Similar to the toy task given above, we can apply the method to the case task by setting up $\Delta\mathbb{P}$ and $\mathbb{C}$ matrices according to the actual parameters. For example, the Finnish NER dataset requires $K$ and $|M|$ to be set to 10 and 89, respectively. These values are 3 and 181, respectively, for the Turkish NER dataset. The method provides the explanation vectors $\mathbf{e}_{ij}^{k}$ for every entity tag $k$ in every region $r_{ij}$ for every sentence $X_i$. The values in the $m^{\text{th}}$ dimension of this vector are predictions on how much and in which direction $m^{\text{th}}$ morphological tag impacts the prediction of entity tag $k$ in region $r_{ij}$.

**Algorithm 1** The explanation method for sequence-based NLP tasks. `predict` function relies on the model to obtain the probabilities for each label $k$.

$X \leftarrow$ set of sentences to be explained
$K \leftarrow$ number of classes
**for** $i = 1$ to $|X|$ **do**
 **for all** region $j$ in sentence $X_i$ **do**
 $\mathbb{F}_{ij} \leftarrow$ set of features in $r_{ij}$
 $\pi_{ij} = \{\texttt{remove}(X_i, j, f) : f \in \mathbb{F}_{ij}\} \cup \{X_i\}$
 **for all** perturbed sentence $\pi_{ij}^{r}$ in $\pi_{ij}$ **do**
 $p_{ij}^{r} \leftarrow$ vector of probabilities of all labels using $\texttt{predict}(\pi_{ij}^{r})$
 $\Delta\mathbb{P}_{ij} \leftarrow$ filled such that $r^{\text{th}}$ column is $p_{ij}^{r} - p_{ij}^{\emptyset}$ and the last column is $\vec{0}$
 $\mathbb{C}_{ij}(r,:) \leftarrow$ vector of zeros, minus ones, and ones representing the perturbed sentence $\pi_{ij}^{r}$

**for all** label $k$ **do**
$$\mathbf{e}_{ij}^{k\star} = \text{argmin} ||\Delta\mathbb{P}_{ij}(k,:) - \mathbb{C}_{ij}e_{ij}^k||_2^2 + ||e_{ij}^k||_2^2$$

## Analysis

To assess the proposed method, two NER taggers were trained for Turkish and Finnish with appropriate datasets [7]. A standard training regime was followed as in the paper that introduced the NER tagger model [7]. However, 100 dimensional emdeddings were used instead of 10.

As the NER tagger jointly models the NER task and the morphological disambiguation (MD) task, two data sources are required for each language (Table 3). For Finnish, we used a NER dataset of 15436 sentences for training NER related parts of the model [31, 32]. For MD related parts, we used 172,788 sentences from the Universal Dependencies dataset [13] using the modified version [33] of the UdPipe morphological tagger [34] to obtain all possible morphological analyses instead of the most probable morphological analyses provided by the original version.

For Turkish, we used the most prevalent NER dataset which includes the correct MD labels, but, unfortunately, omits the other morphological analyses required for training the model. The dataset which includes all morphological analyses was obtained from the repository associated with the article which introduced the NER tagger [7].

## Results

The evaluation of explanation methods remains an active research area. Several approaches have emerged ranging from manual assessments to qualitative and qualitative methods with no consensus as of yet [35]. To evaluate the explanation vectors, we utilize three metrics based on the mean of standardized importance values ($\hat{\mu}_k$), the distribution of standardized importance values across the corpus ($\hat{\mathbb{E}}^k(m)$), and the mutual information gain ($MI_{k,m}$), which are defined in the next section.

We evaluate the explanations as follows:

1. As a form of qualitative evaluation, the average importance values for each morphological tag $m$ (denoted as $\hat{\mu}_k(m)$) are ranked in order of significance. This ranking is compared with the expected ranking based on our knowledge of the language features.

2. We visually inspect the importance values of the morphological tags using $\hat{\mathbb{E}}^k(m)$.

3. We determine the morphological tags that are important for all entity tags using $\hat{\mu}_k(m)$.

4. As a quantitative approach, we calculate the mutual information gain between each morphological tag ($m$) and entity tag ($k$) denoted as $MI_{k,m}$ and rank the morphological tags according to this metric to observe the number of matches with the results of the proposed method.

**Table 3. The hyperparameters and region-related statistics for Turkish and Finnish NER datasets.**

| Lang. | $M$ | $K$ | # regions | $r$ per sentence | ORG | TIT | PER | TIM | LOC | DATE | PRO | MISC | EVENT | OUTSIDE |
|---|---|---|---|---|---|---|---|---|---|---|---|---|---|---|
| Turkish | 181 | 3 | 22364 | 1.85 | 6268 | - | 9318 | - | 6778 | - | - | - | - | - |
| Finnish | 89 | 10 | 25578 | 2.27 | 9101 | 631 | 2229 | 5148 | 2040 | 956 | 4467 | 908 | 93 | 5 |

5. Finally, we designed an experiment to observe whether removing the higher ranked morphological tags more significantly decreases the performance in comparison to the removal of lower ranked tags.

These approaches are used to evaluate the computed explanation vectors for Turkish and Finnish. The code for computing the metrics are shared with the research community on a public website [9].

## Metrics

Two separate NER models are trained for Finnish and Turkish to demonstrate the proposed method. We process the corpora so that $F_{it}$ is the union of all morphological tags in all possible morphological analyses of the $t^{\text{th}}$ word in the $i^{\text{th}}$ sentence. We then employ the explanation method given in Algorithm 1 to obtain the explanation vectors $\mathbf{e}_{ij}^{k\star}$ of size $|M|$ for every named entity region $r_{ij}$ in every sentence $X_i$ by solving Eq 2.

We then calculate

$$\mu_k = \frac{\sum_{i,j} \mathbf{e}_{ij}^{k\star}}{N_k}, \tag{6a}$$

$$\sigma_k = \sqrt{\frac{\sum_{i,j} \left(\mathbf{e}_{ij}^{k\star} - \mu_k\right)^2}{N_k}}, \tag{6b}$$

$$\hat{\mathbf{e}}_{ij}^{k\star} = \frac{\mathbf{e}_{ij}^{k\star} - \mu_k}{\sigma_k} \tag{6c}$$

$$\hat{\mu}_k = \frac{\sum_{i,j} \hat{\mathbf{e}}_{ij}^{k\star}}{N_k}. \tag{6d}$$

where $N_k$ is the number of named entity regions labeled with the named entity $k$ in the corpus. The $\mu_k$ and $\sigma_k$ are vectors of size $|M|$ where each dimension is the mean and variance of the importance values of the corresponding morphological tag $m$. The $\hat{\mathbf{e}}_{ij}^{k\star}$ is obtained by standardizing the values using the mean and variance of $\mathbf{e}_{ij}^{k\star}$. The $\hat{\mu}_k$ is the standardized version of $\mu_k$.

Furthermore, we define

$$\hat{\mathbb{E}}^k(m) = [\,\hat{\mathbf{e}}_{ij}^{k\star}(m) : \forall i,j\,] \tag{7}$$

as a vector containing all values in the $m^{\text{th}}$ dimension of all explanation vectors in all regions $r_{ij}$ with label $k$. This variable is useful in analyzing the distribution of standardized importance values across the corpus.

The metric $MI_{k,m}$ is defined to quantify the information given by a morphological tag $m$ to predict an entity tag $k$ in any given region. To calculate this metric, a pair of vectors $(L_k, \Phi_{k,m})$ are defined. Each vector has N dimensions which correspond to the total number of regions in the corpus. Each dimension in $L_k$ is set to 1 if the region is labeled with entity tag $k$, otherwise to 0. Likewise, each dimension of $\Phi_{k,m}$ is set to 1 if the region contains morphological tag $m$, otherwise to 0. Using these vectors, the mutual information score is computed for each pair of

$k$ and $m$:

$$MI_{k,m} = \sum_{j=0}^{1}\sum_{j'=0}^{1} \frac{|L_k(j) \cap \Phi_{k,m}(j')|}{N} log \frac{N|L_k(j) \cap \Phi_{k,m}(j')|}{|L_k(j)||\Phi_{k,m}(j')|} \tag{8}$$

where $L_k(j)$ and $\Phi_{k,m}(j)$ denote the set of indices that are set to $j$.

## Using standardized mean importance values

To assess the importance of a morphological tag $m$ for predicting entity tag $k$ across the corpus, we examine the $m^{\text{th}}$ dimension of $\hat{\mu}_k$. We chose this approach instead of assigning higher importance to features which are used to explain more instances throughout the corpus as in the original LIME approach [6]. Our approach avoids falsely marking very common features as important. For instance, the morphological tag 'Case = Nom' which indicates the nominal case can be found in many words for all entity tags. If we were to assign high importance according to the frequency across the corpus, we would incorrectly declare this type of features as important. Using standardized mean importance values in $\hat{\mu}_k$ is better in this regard.

We rank the morphological tags ($m$) for each entity tag $k$ using $\hat{\mu}_k(m)$. The ranked morphological tags for Finnish and Turkish can be seen in Tables 4 and 7, respectively. To conserve space, these tables show only the tags that appear in the top 10 (for Finnish) and top 20 (for Turkish). Rows in these tables are morphological tags ($m$) and columns are entity tags ($k$). Each cell gives the rank of the corresponding $\hat{\mu}_k(m)$. A high rank (one being the highest) indicates a positive relation, whereas a low rank indicates a negative relation with respect to the prediction of entity tag $k$.

**Finnish.** The five morphological tags with the highest and lowest ranks in a column exhibit a coherent picture. The highest ones are generally related to the entity tag, while the lowest ones are either unrelated or are in contradiction with the semantics of the entity tag. For example, for Finnish, the first seven morphological tags in column 'LOC' of Table 4 include five case related tags which indicate the inessive 'Case = Ine', genitive 'Case = Gen', elative 'Case = Ela', illative 'Case = Ill', and adessive 'Case = Ade' cases. All of these cases are related to the locative semantics of the attached word. The column for 'TIM' also shows a similar relation. The essive case marker 'Case = Ess' which is related with temporal semantics is in the top three morphological markers of the Finnish 'TIM' entity. Additionally, we observe that the illative 'Case = Ill' and innessive 'Case = Ine' cases are among the most negative ones ($87^{\text{th}}$ and $89^{\text{th}}$) in the column of the 'DATE' entity tag. This is expected because these are known to be related to location expressions. This kind of observations can help in assessing the quality of a trained model.

In addition to this analysis, we evaluated the importance of the tags with another approach where we exploited the rule-based NER tagger which was used to validate the Finnish dataset [31, 32].

The Finnish dataset was created with manual annotations, which were subsequently validated with the rule-based FINER NER tagger to improve the quality [36]. FINER is a part of the `finnish-tagtools` toolkit which contains a morphological analyzer, a tokenizer, a POS tagger, and a NER tagger for Finnish [36]. The rules of FINER have been specified by linguists and tested on the dataset. As such, the labels produced by them may be considered as gold labels.

The FINER authors define a rule for each named entity tag that matches every instance of it. To tag a sentence, it is tokenized and the morphological tags of every word are determined using a Finnish morphological analyzer. These morphological tags are then disambiguated using a POS tagger. The output of these tools consists of the surface form, the lemma, the

**Table 4. The ranks of $\hat{\mu}_k(m)$ for each entity tag in Finnish.**

| Morphological Tag | ORG | TIT | PER | TIM | LOC | DATE | PRO | MISC | EVENT | OUTSIDE | median | std. dev. |
|---|---|---|---|---|---|---|---|---|---|---|---|---|
| Number = Sing | 89 | **1** | 89 | 89 | 89 | 88 | 89 | **2** | **3** | 89 | 89 | 42 |
| Case = Nom | 88 | 89 | **3** | 82 | 88 | 11 | **1** | 89 | **1** | **1** | 3 | 39 |
| Voice = Act | **3** | 83 | **5** | 80 | 79 | **6** | 12 | 83 | 84 | 85 | 12 | 30 |
| VerbForm = Fin | **6** | 79 | **9** | 61 | **8** | 12 | **8** | 78 | 12 | 86 | 8 | 26 |
| Mood = Ind | **8** | 81 | 70 | 52 | **9** | 14 | **9** | 81 | 16 | 88 | 9 | 27 |
| Number = Plur | **4** | 85 | **4** | 86 | **3** | 23 | 78 | **1** | 89 | 75 | 4 | 35 |
| Case = Gen | 87 | 87 | 88 | 79 | **2** | **7** | 88 | 13 | 87 | **2** | 87 | 37 |
| Degree = Pos | **2** | 88 | 85 | **2** | 87 | **1** | 82 | 84 | **6** | **3** | 82 | 41 |
| Person = 3 | 20 | 84 | **7** | 54 | 17 | 33 | 76 | 66 | 28 | 83 | 20 | 19 |
| Tense = Pres | 10 | **9** | 16 | 68 | 10 | 13 | 14 | 76 | 14 | 87 | 14 | 32 |
| Case = Par | 17 | 86 | 87 | 85 | **6** | 86 | **2** | 85 | **7** | 14 | 7 | 28 |
| Style = Coll | 14 | 74 | **2** | **9** | 21 | 32 | 16 | 50 | 32 | 22 | 16 | 22 |
| VerbForm = Part | 23 | 65 | 57 | 76 | 39 | **3** | 33 | 73 | **8** | 36 | 33 | 27 |
| Case = Ine | 84 | 15 | 24 | **4** | **1** | 89 | **5** | 87 | 85 | 35 | 24 | 35 |
| Case = Ill | 33 | 14 | 77 | 75 | **5** | 87 | **3** | 82 | 31 | 34 | 31 | 29 |
| PartForm = Past | 25 | 69 | 60 | 77 | 74 | **5** | 29 | 65 | 10 | 33 | 29 | 26 |
| Case = Ela | 83 | 62 | 68 | 67 | **4** | 19 | **4** | 86 | 42 | 32 | 42 | 24 |
| VerbForm = Inf | 13 | 31 | 21 | 49 | 84 | **8** | 84 | 12 | 86 | 77 | 84 | 25 |
| Voice = Pass | 27 | 70 | 56 | 55 | 19 | **4** | 79 | 71 | 27 | 31 | 27 | 25 |
| Person = 1 | **9** | **8** | 73 | 69 | 11 | 26 | **6** | 75 | 29 | 78 | 11 | 28 |
| Case = Ade | 86 | 37 | 69 | 66 | **7** | 30 | 11 | 80 | 82 | **4** | 69 | 26 |
| Connegative = Yes | 22 | 63 | 20 | **7** | 26 | 15 | 23 | **6** | 17 | 84 | 22 | 32 |
| Person[psor] = 3 | 55 | 34 | 38 | 60 | 76 | 43 | 18 | **3** | 81 | 26 | 55 | 18 |
| Case = All | 54 | 75 | 81 | 65 | 12 | 36 | 13 | 79 | **2** | 23 | 13 | 22 |
| Derivation = Minen | 31 | **3** | 83 | 83 | 22 | 29 | 15 | **7** | 19 | 20 | 22 | 28 |
| NumType = Card | 16 | 68 | 30 | 88 | 82 | 21 | 87 | 77 | **5** | **6** | 30 | 32 |
| Case = Ess | 35 | 28 | 55 | **3** | 13 | **2** | 26 | 63 | 34 | **7** | 34 | 23 |
| Case = Tra | 53 | 12 | 17 | 51 | 70 | 28 | 43 | **9** | 38 | **8** | 43 | 16 |
| Typo = Yes | 28 | 76 | 64 | **5** | 83 | 24 | 40 | 64 | 13 | **9** | 40 | 29 |
| Mood = Imp | 34 | 23 | **8** | **8** | 25 | 17 | 75 | 16 | 23 | **5** | 25 | 6 |
| Abbr = Yes | **7** | **5** | **1** | 22 | 78 | 82 | 85 | 74 | **4** | 82 | 7 | 32 |
| Foreign = Yes | **5** | 11 | 86 | 27 | 23 | 69 | 17 | **5** | 11 | 81 | 17 | 30 |
| Clitic = Kin | 29 | 18 | **6** | **6** | 81 | 79 | **7** | **8** | 33 | 18 | 29 | 27 |
| Case = Ins | 15 | 17 | 79 | 64 | 69 | **9** | 20 | 59 | 24 | 79 | 24 | 27 |
| Derivation = Lainen | 41 | **2** | 78 | 71 | 16 | 71 | 22 | 70 | 18 | 62 | 22 | 26 |
| NumType = Ord | 11 | **6** | 75 | 59 | 33 | 84 | 81 | **4** | **9** | 64 | 33 | 32 |
| Derivation = Inen | 79 | **4** | 71 | 62 | 27 | 34 | 73 | 19 | 21 | 69 | 71 | 24 |
| Derivation = Vs | 48 | **7** | 44 | 74 | 48 | 75 | 47 | 68 | 69 | 46 | 48 | 25 |
| UNKNOWN* | **1** | 77 | 84 | **1** | 86 | 85 | 86 | 88 | 83 | 76 | 84 | 32 |
| Tense = Past | 12 | 80 | 76 | 21 | 15 | 45 | **10** | 72 | 78 | 12 | 15 | 26 |
| PartForm = Pres | 50 | **10** | 23 | 20 | 29 | 25 | 41 | 60 | 43 | 13 | 41 | 17 |
| Mood = Cnd | 77 | 67 | 19 | 23 | 36 | 53 | 39 | 20 | 26 | **10** | 36 | 21 |
| InfForm = 2 | 18 | 38 | 35 | **10** | 85 | **10** | 80 | 11 | 15 | 80 | 35 | 27 |
| Clitic = Han | 80 | 49 | **10** | 16 | 20 | 60 | 54 | 48 | 58 | 59 | 54 | 15 |
| Mood = Pot | 39 | 13 | 32 | 32 | 67 | 58 | 27 | **10** | 56 | 40 | 39 | 17 |

The morphological tags that are in the first 10 for at least one entity tag are shown.

**Table 5. Three FINER rules are presented in the order of increasing generality.**

| | | |
|---|---|---|
| 1 | **Rule** `PropGeoLocInt` | |
| 2 | **Surface form** | Field FSep—i.e. any string is allowed |
| 3 | **Lemma** | Field FSep—i.e. any string is allowed |
| 4 | **Morphological tags** | Field [{NUM = SG} Field {CASE=}[{INE}|{ILL}|{ELA}]] Field FSep |
| 5 | **Extra labels** | Field [{PROP = GEO}] Field FSep |
| 6 | **Rule** `LocGeneral2` | |
| 7 | **Single word** | PropGeoGen |
| 8 | **Right context** | RC(WSep PropGeoLocInt) |
| 9 | **Rule** `Location` | |
| 10 | **Option 1** | Ins(LocGeneral) |
| 11 | **Option 2** | Ins(LocGeogr) |
| 12 | **Option 3** | Ins(LocPolit) |
| 13 | **Option 4** | Ins(LocStreet) |
| 14 | **Option 5** | Ins(LocAstro) |
| 15 | **Option 6** | Ins(LocPlace) |
| 16 | **Option 7** | Ins(LocFictional) |

The first one acts on single words. The second applies a rule on a single word and requires the right context to match another rule. The last one is the top rule for the 'Location' tag. Options 1, 2, 3, and 6 can also be seen in Fig 7 as they lie on paths that reach a morphological tag while other options do not lead to any.

disambiguated morphological tags, and some extra labels such as proper name indicators. The rules for each named entity tag are matched using this output. The first successful match designates the named entity tag. Each rule is a regular expression or a combination of several other rules, using concatenation, union, or intersection operators in `pmatch` syntax [37]. For example, the rule named `PropGeoLocInt` in Table 5 matches a single word only if the word's morphological tag includes 'NUM = SG', one of the three case morphemes 'CASE = INE', 'CASE = ILL', 'CASE = ELA', and the proper noun label 'PROP = GEO' (e.g. Finnish word 'Kiinassa' which means 'in China' matches this rule). In rows 2 and 3 of the table, the simple rules called 'Field' and 'FSep' are used to match a string of any length and a tab character, respectively. In rows 4 and 5 of the table, we see a specific concatenation of these two rules and several string literals in curly brackets. These pieces make up the rule that matches the surface form, the lemma, the disambiguated morphological tags, and the extra labels. However, a successful match of this rule does not result in a named entity tag. Instead, it is used in more general rules as seen in the definition of `LocGeneral2` in rows 7-8 of Table 5. It matches the rule `PropGeoGen` and the rule `PropGeoLocInt` in the right context. This hierarchical structure continues up to the top rule for the 'Location' named entity tag, namely the `Location` rule. The morphological tags used in FINER are different from the ones in our paper as it employs the `Omorfi` morphological analyzer. However, the mapping is straightforward except for a few cases such as 'VerbForm = Fin' and 'Mood = Ind' [38].

We form a graph to evaluate the overlap between the morphological tags output by our method and those specified in the FINER rules. The internal nodes in the graph correspond to the rule names and the leaf nodes to the morphological tags. We define an edge from rule A to rule B if and only if the definition of rule A contains a reference to rule B in the form of concatenation, union or intersection operators. We process this graph to produce a subgraph for each of the nodes that correspond to named entity tags. For this, we start from the node of a named entity tag and traverse the graph breadth-first for a maximum of four iterations.

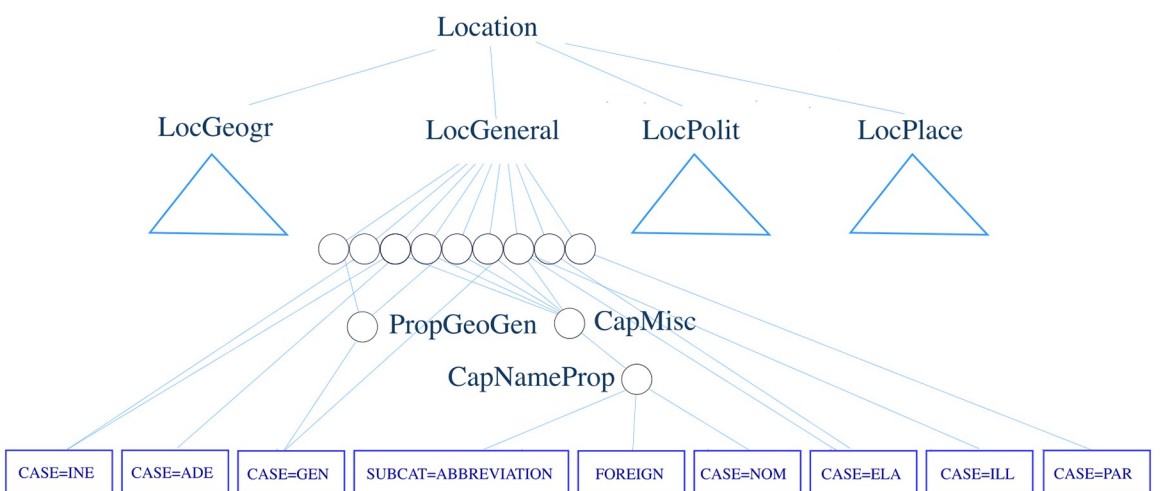

**Fig 7. A subset of the graph that is formed based on the FINER rules.** It shows the nodes and the connections that lie between the 'Location' top rule and the morphological tags reachable using the rules. We only show the subgraph reachable from 'LocGeneral' in finer detail, while replacing the other subgraphs with triangles. Morphological tag nodes are represented with a rectangle. Circles represent the internal rules.

Although the resulting subgraph is not strictly a tree, it is highly hierarchical. In Fig 7, we present a subgraph for 'Location' to demonstrate the hierarchical nature of the FINER rules graph. In the figure, 'PropGeoGen' is referred to by two rules, which themselves are referred to by 'LocGeneral'. One of them is 'LocGeneral2' and the other is 'LocGeneralColloc1'. In a subgraph, we follow every possible path from the root node to the leaf nodes that are composed of the `Omorfi` equivalents of the morphological tags from Table 4. If there is at least one such path, we assume that the morphological tag is related to this named entity tag and thus may be used to evaluate the list of important tags produced by our method.

Table 6 shows the matching results between the proposed explanation method and the FINER tagger for the five most-frequently occurring (more than 1000 occurrences) named entity tags in the dataset. We ignored the morphological tags that are not in the top 10 of the importance list of the proposed explanation method. This resulted in a set of 19 morphological tags which was used during the following evaluation. We quantify the rate of matching by counting the number of successful decisions by our method. When it is concurrently true that a morphological tag is predicted as important for a named entity tag and there is at least one path from the named entity to the morphological tag, we regard this as a true positive (TP). If it is predicted to be important by our method but no paths exist between the named entity tag and the morphological tag, it is counted as a false positive (FP). For example, all predictions for 'Location' are correct, i.e. all of our predicted morphological tags are reachable from the

**Table 6. Results of the matching between the proposed explanation method and the FINER tagger for the five most-frequently occurring named entity tags.**

|  | TP | TN | FP | FN | Precision % | Recall % | F1-measure % |
|---|---|---|---|---|---|---|---|
| Location | 7 | 7 | 0 | 5 | 100.00 | 58.33 | 73.68 |
| Organization | 4 | 8 | 1 | 6 | 80.00 | 40.00 | 53.33 |
| Person | 4 | 8 | 3 | 4 | 57.14 | 50.00 | 53.33 |
| Product | 2 | 7 | 4 | 6 | 33.33 | 25.00 | 28.57 |
| Time | 1 | 9 | 7 | 2 | 12.50 | 33.33 | 18.18 |
| Total | 18 | 49 | 22 | 25 | 45.00 | 41.86 | 43.37 |

named entity tag. However, five morphological tags that are predicted as unimportant to 'Location' have paths originating from the 'Location' named entity tag. These are regarded as false negative (FN) predictions. Other seven predictions are counted as true negatives (TN).

In Table 6, we see a mostly assuring picture. The precision rates of 'Location' and 'Organization' are quite high while that of the 'Person' type is lower. This is due to the fact that two of the three false positives in 'Person' ('Style = Coll' and 'Person = 3') are absent in the FɪNER rules. The first one is a tag specific to this dataset and the second one does not exist in FɪNER rules although it is a tag found in our corpus. The false positive in 'Organization' is also the 'Person = 3' tag. The worst recall rate occurs with our predictions for the 'Product' named entity tag. The recall ratio indicates that we miss about 75% of the morphological features which are important according to the FɪNER rules. Some of the missed ones are among the most common morphological tags such as 'Case = Gen' and 'Number = Plur', which are used in many basic rules such as 'PropGeoGen'. These basic rules appear in many paths that start from any named entity tag. This inevitably results in many paths that end at these morphological tags for each named entity tag, thus lowering the recall rate for all. The same observation is valid for the other four named entity tags also in the sense that these common tags are included within their false negatives.

**Turkish.** An inspection of the 'ORG', 'LOC', and 'PER' entity tag columns in Table 7 for Turkish reveals that the tag that indicates proper nouns ('Prop') is the dominant one. This shows that the model relies on the morphological analyzer's performance to mark proper nouns correctly. However, the case of 'P3sg' is more interesting. This morpheme is commonly found in noun clauses which are organization or location names. On the other hand, it is never attached to person names. The case of 'P3pl' is similar. This is reflected in our results; these morphological tags are not positively related with the 'PER' entity tag as seen in the table. The case of 'Almost' is interesting as it is a rare morpheme and is almost never attached to the correct morphological analysis. These properties should have made it an unimportant tag. On the contrary, it is regarded as an important tag for 'ORG' and 'LOC' named entity tags by our method. One possible explanation is that when 'Almost' is removed from the feature sets to create perturbed sentences, the morphological analyses that contained 'Almost' before perturbation is regarded more probable by the tagger, which in turn decreases the probability of the prediction. This causes the morpheme 'Almost' to be considered as an important morphological tag with explanatory value.

## Importance distributions of morphological tags

The histograms of importance values for various combinations of entity types and morphological tags are shown in Fig 8 using the Finnish NER corpus. Each row in Fig 8a is a heatmap that represents a histogram of values in $\hat{\mathbb{E}}^{\text{PER}}(m)$, which is a vector that consists of all importance values of the morphological tag $m$ explaining a 'PER' entity tag prediction in a region. Bin edge positions are determined by $-10^{i/10}$ and $10^{i/10}$ for the negative and positive sides, respectively, where $i \in \{-25, \ldots, 13\}$. The frequency corresponding to each bin is coded with color tones from white to black. The morphological tags are ordered in descending order from top to bottom by their mutual information gain $MI_{\text{PER},m}$. However only the first 20 morphological tags are shown here due to space constraints. Fig 8b is formed likewise. There is often a clustering between −0.050 and 0.063 when all morphological tags are considered. This clustering vanishes when only the first 20 morphological tags are considered. We argue that this is correlated with high mutual information gain values corresponding to higher ranked morphological tags.

**Table 7. The ranks of $\hat{\mu}_k(m)$ for each entity tag in Turkish.**

| Morphological Tag | ORG | LOC | PER | median | std. dev. |
|---|---|---|---|---|---|
| Noun | 176 | 181 | **2** | 89 | 0 |
| A3sg | **2** | 180 | 178 | 90 | 0 |
| Verb | **8** | 147 | 173 | 90 | 0 |
| Adj | **3** | 170 | 176 | 89 | 0 |
| P3sg | **4** | **2** | 169 | 86 | 0 |
| Pos^DB | **17** | 160 | 171 | 94 | 0 |
| Punc | **20** | **5** | **7** | 13 | 0 |
| Pos | **13** | **16** | **4** | 8 | 0 |
| Prop | **1** | **1** | **1** | 1 | 0 |
| Acc | **12** | 169 | 174 | 93 | 0 |
| Nom^DB | **9** | 166 | 170 | 89 | 0 |
| Loc | **19** | 177 | 161 | 90 | 0 |
| Adverb | 46 | **14** | **19** | 32 | 0 |
| Conj | 28 | **6** | **18** | 23 | 0 |
| Num | 21 | **13** | **9** | 15 | 0 |
| Pass | 59 | **19** | **17** | 38 | 0 |
| P3pl | **14** | **12** | 70 | 42 | 0 |
| Adj^DB | **15** | 172 | 156 | 85 | 0 |
| Past | 62 | 153 | **14** | 38 | 0 |
| Card | 36 | **18** | **16** | 26 | 0 |
| Imp | **16** | **7** | **6** | 11 | 0 |
| A2sg | **18** | **8** | **5** | 11 | 0 |
| Ness | **11** | 150 | 147 | 79 | 0 |
| Aor^DB | 32 | **17** | 164 | 98 | 0 |
| Agt | 22 | **9** | 172 | 97 | 0 |
| With | 27 | 158 | **10** | 18 | 0 |
| P1sg | 24 | 30 | **8** | 16 | 0 |
| Almost | **6** | **4** | 139 | 72 | 0 |
| A2pl | 41 | 28 | **11** | 26 | 0 |
| Opt | 51 | 29 | **13** | 32 | 0 |
| Noun^DB | **5** | **3** | 138 | 71 | 0 |
| Equ | 37 | **15** | **15** | 26 | 0 |
| Ord | 26 | **11** | **12** | 19 | 0 |
| UNKNOWN* | **10** | 165 | **3** | 6 | 0 |
| Pron | 56 | 55 | **20** | 38 | 0 |
| Interj | 173 | **20** | 32 | 102 | 0 |

The morphological tags that are in the first 20 for at least one entity tag are shown.

Fig 8c and 8d show that the explanation values for the morphological tags 'Case = Ine' and 'Case = Nom' are distributed in a different way by plotting the histograms of $\hat{\mathbb{E}}^{\text{LOC}}(m)$ where $m$ is the corresponding dimension. These figures have the same $x$ axis as in Fig 8a and 8b. We should note that the $x$ axis is in log-space so that the clusters near the center are very close to zero, whereas the concentration around 7.94 indicates that a significant portion of the importance values are high.

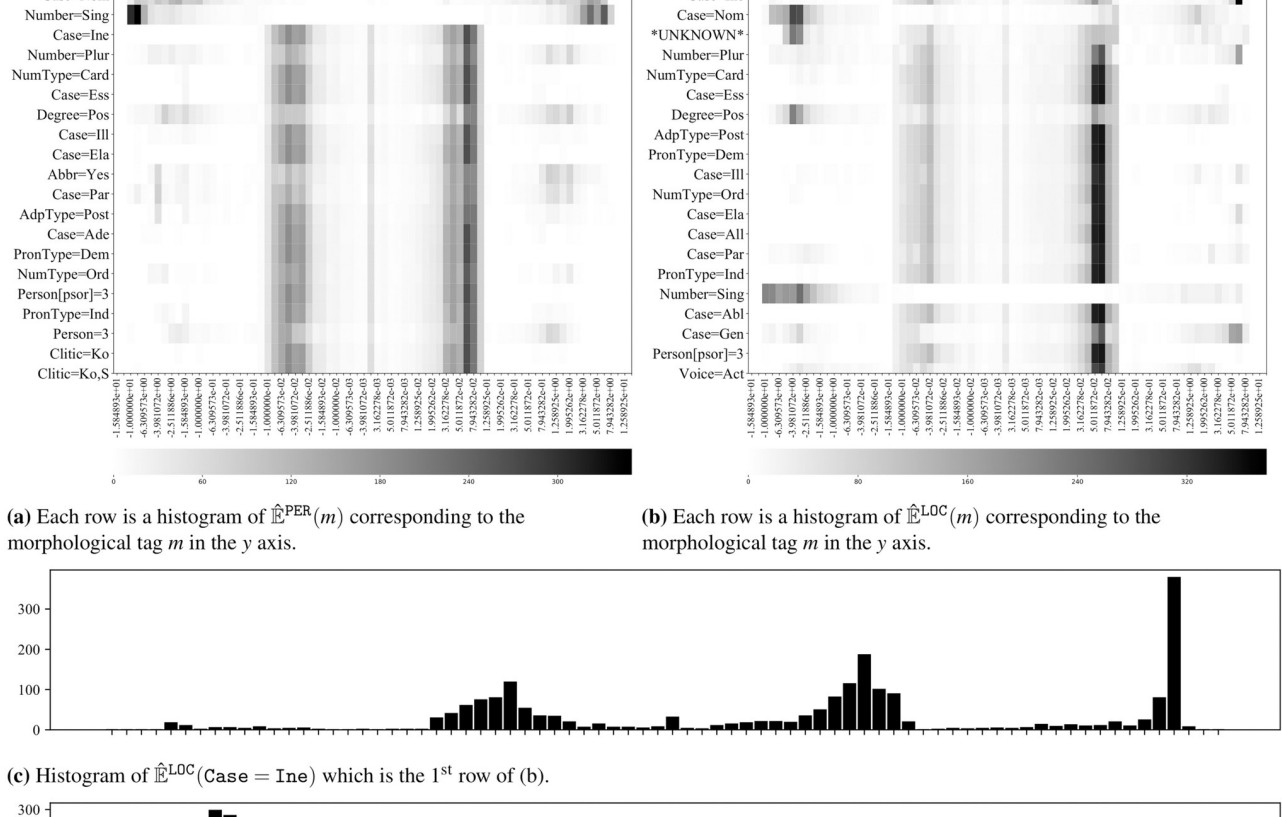

**(a)** Each row is a histogram of $\hat{\mathbb{E}}^{\text{PER}}(m)$ corresponding to the morphological tag $m$ in the $y$ axis.

**(b)** Each row is a histogram of $\hat{\mathbb{E}}^{\text{LOC}}(m)$ corresponding to the morphological tag $m$ in the $y$ axis.

**(c)** Histogram of $\hat{\mathbb{E}}^{\text{LOC}}(\texttt{Case} = \texttt{Ine})$ which is the 1$^{\text{st}}$ row of (b).

**(d)** Histogram of $\hat{\mathbb{E}}^{\text{LOC}}(\texttt{Case} = \texttt{Nom})$ which is the 2$^{\text{nd}}$ row of (b).

**Fig 8. The histograms of importance values for various entity tag and morphological tag combinations.** Darker colors indicate higher frequencies. Calculated over the training corpus in Finnish. All histograms in the figures share the same bin edges. Edge positions are calculated by $-10^{i/10}$ and $10^{i/10}$ for the negative and positive sides, respectively, where $i \in \{-25, \ldots, 13\}$. (**a**) Each row is a heatmap that represents a histogram of $\hat{\mathbb{E}}^{\text{PER}}(m)$ corresponding to the morphological tag $m$. (**b**) Each row is a heatmap that represents a histogram of $\hat{\mathbb{E}}^{\text{LOC}}(m)$ corresponding to the morphological tag $m$. (**c**) The histogram of $\hat{\mathbb{E}}^{\text{LOC}}(\texttt{Case} = \texttt{Ine})$ which is the 1$^{\text{st}}$ row of (b). (**d**) The histogram of $\hat{\mathbb{E}}^{\text{LOC}}(\texttt{Case} = \texttt{Nom})$ which is the 2$^{\text{nd}}$ row of (b).

### Importance of morphological tags across the entity tags

To determine the morphological tags that are important across entity tags, we count the number of times the rank of $\hat{\mu}_k(m)$ is in the top or bottom 10 ranks. The morphological tags are sorted by the sum of these frequencies for the features that are ranked at the top and bottom of the list. The first 10 morphological tags with the highest sum for Finnish are shown in Table 8 which are the most frequently encountered tags for most languages. They signify singular or plural, active or passive, and mark the word as nominal, genitive, or inessive cases.

**Table 8. The frequency of ranks that are in the first ten of an entity tag for each morphological tag in Finnish.**

| Morphological Tag | Top | Bottom | Top+Bottom |
|---|---|---|---|
| Number = Sing | 3 | 7 | 10 |
| Degree = Pos | 5 | 5 | 10 |
| Case = Nom | 4 | 5 | 9 |
| Case = Gen | 3 | 5 | 8 |
| Case = Par | 3 | 5 | 8 |
| UNKNOWN* | 2 | 6 | 8 |
| Number = Plur | 4 | 3 | 7 |
| Voice = Act | 3 | 4 | 7 |
| Abbr = Yes | 4 | 3 | 7 |
| Case = Ine | 3 | 4 | 7 |

## Quantitative validation using mutual information

In order to validate the explanations created by the proposed explanation model, we employ $\hat{\mu}_k$ (Eq 6d) and $MI_{k,m}$ (Eq 8). We denote the 10 morphological tags ($m$) with the highest $\hat{\mu}_k(m)$ values as $\mathbb{I}_k$. Independently, we calculate the mutual information gain $MI_{k,m}$ between the probability of each morphological tag $m$ being in region $r_{ij}$ and the probability of entity tag $k$ being the label of region $r_{ij}$. We call the first 10 morphological tags with the highest mutual information score as $\mathbb{J}_k$.

$\mathbb{I}_k$ represents the proposed method's list of globally important morphological tags, whereas $\mathbb{J}_k$ is a list created by information gain independent of any particular model. The degree of agreement between these lists gives a quantifiable metric to evaluate different explanation methods. We proceed to take the intersection of $\mathbb{I}_k$ and $\mathbb{J}_k$ for each entity tag $k$ and report the common morphological tags in Table 9 for Finnish and Turkish. The number of morphological tags that are both in $\mathbb{I}_k$ and $\mathbb{J}_k$ hints that the proposed explanation method can correctly predict the morphological tags with high information gain.

## Effect of the absence of a morphological tag

After the rankings of the morphological tags using the average importance values $\hat{\mu}_k(m)$ are obtained for each morphological tag $m$ and entity tag $k$ (Tables 4 and 7), we considered ways of using this knowledge to obtain an improved version of our model. One idea was to modify the architecture of the NER tagger so that it pays more attention to the higher ranked tags compared to the other tags. For example, an extra dimension in the morphological tag embedding to represent the rank of the corresponding morphological tag can be exploited by the neural network. However, as the results of the explanation method are relevant only in the context of the specific model that is being inspected, this would result in a new model related to the original model. If the original model was successful in exploiting the morphological features that are really important to the NER task, this approach would yield successful. On the other hand, if the original model was not able to exploit the important morphological tags due to the training regime or the inefficiency of the architecture, our method would falsely indicate other morphological tags instead of the important tags. This approach would yield a model with less performance. So, training a model which exploits the higher ranked morphological tags reported in our study might not result in an improved performance.

Instead, we decided to test the hypothesis that higher ranked morphological tags can improve the performance for NER by following a corpus-based approach. For the 'Location'

**Table 9. Common Finnish and Turkish morphological tags that are both in $\mathbb{I}_k$ and $\mathbb{J}_k$.**

| Finnish | | |
|---|---|---|
| Entity tag | Morphological tags | Agreement rate |
| ORG | 'Case = Gen', 'Number = Plur', 'Number = Sing', 'Case = Nom', 'Degree = Pos', '*UNKNOWN*' | 0.6 |
| TIT | 'Case = Gen', 'Case = Par', 'Number = Sing', 'Case = Nom' | 0.4 |
| PER | 'Number = Plur', 'Number = Sing', 'Case = Nom', 'Abbr = Yes', 'Degree = Pos' | 0.5 |
| TIM | 'Number = Sing', 'Case = Ess', 'Degree = Pos', 'Case = Par', 'PronType = Dem', '*UNKNOWN*' | 0.6 |
| LOC | 'Number = Plur', 'Case = Nom', 'Degree = Pos', 'Case = Ill', 'Case = Ine', '*UNKNOWN*' | 0.6 |
| DATE | 'Case = Ess', 'Case = Ine', 'Degree = Pos', '*UNKNOWN*' | 0.4 |
| PRO | 'Case = Par', 'Case = Nom', 'Case = Ela', 'NumType = Card' | 0.4 |
| MISC | 'Number = Plur', 'Number = Sing', 'Case = Ela', 'Person[psor] = 3', 'Degree = Pos', 'Case = Par', '*UNKNOWN*' | 0.7 |
| EVENT | 'Case = Ine', 'Number = Plur', 'Voice = Act' | 0.3 |
| OUTSIDE | 'Case = Ade', 'Number = Sing', 'Case = Nom', 'Degree = Pos' | 0.4 |
| Turkish | | |
| Entity tag | Morphological tags | Agreement rate |
| ORG | 'Adj', 'P3sg', 'Prop', 'A3pl', 'Nom' | 0.5 |
| LOC | 'Loc', 'Dat', 'P3sg', 'Prop', 'Nom', 'Gen' | 0.6 |
| PER | 'Adj', 'Gen', 'Dat', 'Nom' | 0.4 |

named entity tag in Finnish, we chose the top two ranked morphological tags ('related tags') and eight other randomly selected morphological tags which are not among the first 10 or last 10 ranks ('unrelated tags'). For each of these 10 tags, we created a modified version of the dataset so that no morphological analysis contains the corresponding tag. We then trained and evaluated each of the 10 models separately in two independent runs. We calculated the averages of the F-measure, precision, and recall metrics for the 'Location' named entity tag using these two runs. Table 10 compares the performance of the 'related tags' ('Case = Ine' and 'Case = Gen') and the 'unrelated tags' in terms of the differences in these metrics. For each combination of two 'related' and 'unrelated' tags, we subtract the success rate of the model in which a related tag is removed from the success rate of the model in which an unrelated tag is removed. The average, minimum, and maximum values of the resulting eight difference values are shown in the respective columns. A positive difference in the average column indicates that the removal of the unrelated tags decreases the performance of the model less than the removal of the related tag, while a negative difference indicates the opposite.

**Table 10. Comparison between each 'related tag' and 'unrelated tag' sets using F-measure and Recall metrics.**

| Related tag | Metric | Unrelated tags | | |
|---|---|---|---|---|
| | | Average % Difference | Min % Difference | Max % Difference |
| **Case = Ine** | Precision | 0.31 | -1.83 | 2.86 |
| | Recall | -2.73 | -5.44 | -0.39 |
| | F-measure | -1.31 | -3.07 | -0.21 |
| **Case = Gen** | Precision | -2.34 | -4.48 | 0.21 |
| | Recall | 0.14 | -2.57 | 2.48 |
| | F-measure | -1.07 | -2.83 | 0.03 |

The results shown in Table 10 are contrary to our expectations. Our hypothesis that the related tags contain a stronger signal for the named entity tag and their absence would decrease the model performance is not verified. This is verified only for the precision metric of the 'Case = Ine' tag and the recall metric of the 'Case = Gen' tag. An observation might explain the failure to reject the null hypothesis. An inspection of the morphological analyses reveals that 'unrelated tags' occur less frequently than 'related tags'. We know that there are some co-occurring morphological tags for each tag and the removal of a tag might be compensated by the co-occurring tags. However, this mechanism might not work for 'unrelated tags' as they have relatively fewer co-occurring morphological tags. This might in turn result in a higher loss of performance when an 'unrelated tag' is removed.

## Conclusions and future directions

In this work, we introduced an explanation method which can be employed for any sequence-based NLP task. We introduce the terminology and a procedure which can be adopted to any model that implements a sequence-based NLP task and can transform the model's predictions to be explained with the proposed explanation method. The case study using a joint NER and MD tagger shows that the proposed method can be employed to provide explanations for single input samples to assess the contribution of features to the prediction. Furthermore, it is shown that an analysis of these explanations across the corpus can be helpful in assessing the plausibility of a given trained model.

While forming explanations, we treat each feature in a given region as independent from each other. However, the features may be related to each other in many ways. Firstly, some morphological tags in a single morphological analysis of a given word are dependent on each other. For instance, the presence of one tag may strongly signal the presence of another tag or the order of appearance in the morpheme sequence may be important. Secondly, this dependence may be observed between features inside and outside the region. For example, named entities are usually related to the features of the words to the left or to the right of the context, such as the morphological tags and the characters of the surface forms. In future work, we aim to consider such relationsby extending our model to permit perturbation across multiple regions of variable sizes. The dependency between features can be explored better if our method allows perturbing one or more features either from the region on the left, the region on the right or the region of the named entity itself.

## Author Contributions

**Conceptualization:** Onur Güngör.

**Data curation:** Onur Güngör.

**Investigation:** Onur Güngör.

**Methodology:** Onur Güngör.

**Resources:** Onur Güngör.

**Software:** Onur Güngör.

**Validation:** Onur Güngör.

**Visualization:** Onur Güngör.

**Writing – original draft:** Onur Güngör.

**Writing – review & editing:** Onur Güngör, Tunga Güngör, Suzan Uskudarli.

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
