## [Decision Letter · Decision Letter 0]

1 May 2020

PONE-D-19-26331

EXSEQREG: Explaining sequence-based NLP tasks with regions: With a case study using morphological features for named entity recognition

PLOS ONE

Dear Mr. Gungor,

Thank you for submitting your manuscript to PLOS ONE. After careful consideration, we feel that it has merit but does not fully meet PLOS ONE’s publication criteria as it currently stands. Therefore, we invite you to submit a revised version of the manuscript that addresses the points raised during the review process.

We would appreciate receiving your revised manuscript by Jun 15 2020 11:59PM. To enhance the reproducibility of your results, we recommend that if applicable you deposit your laboratory protocols in protocols.io, where a protocol can be assigned its own identifier (DOI) such that it can be cited independently in the future. For instructions see: http://journals.plos.org/plosone/s/submission-guidelines#loc-laboratory-protocols

We look forward to receiving your revised manuscript.

Kind regards,

Diego Raphael Amancio

Academic Editor

PLOS ONE

2. Please amend your Competing interests statement to declare author commercial affiliations.

Reviewers' comments:

Reviewer's Responses to Questions

**Comments to the Author**

1. Is the manuscript technically sound, and do the data support the conclusions?

Reviewer #1: Yes

Reviewer #2: Yes

2. Has the statistical analysis been performed appropriately and rigorously? 

Reviewer #1: Yes

Reviewer #2: Yes

3. Have the authors made all data underlying the findings in their manuscript fully available?

Reviewer #1: No

Reviewer #2: Yes

4. Is the manuscript presented in an intelligible fashion and written in standard English?

Reviewer #1: Yes

Reviewer #2: Yes

5. Review Comments to the Author

Reviewer #1: 1. The journal paper states that the joint Finnish NER and MD data set is available at the following repository: https://github.com/onurgu/joint-ner-and-md-tagger/tree/master/dataset.

However, at the time of review, no Finnish data set was available at the location indicated by Güngör et al. In the paper, the authors refer to their own previously published paper using an earlier downloadable version of the named entity annotated data set (https://github.com/mpsilfve/finer-data) developed by a reasearch team in Helsinki, Finland. That data set has a PID and is available through FIN-CLARIN at http://urn.fi/urn:nbn:fi:lb-2019050201. The data set has recently been published in the Language Resources and Evaluation Journal as "A Finnish news corpus for named entity recognition" (https://link.springer.com/article/10.1007/s10579-019-09471-7), so a reference to that publication should be added to the list of references as the data set was developed independently of Güngör et al. A post-print of the same journal article is also available at arXiv (https://arxiv.org/abs/1908.04212).

2. The method named EXSEQREG introduces the concept of region that links the prediction and features that are potentially important for the model. A qualitative analysis of the explanations is presented. As the authors point out, there is a lack of established methods for assessing the quality of their predictions.

A region that links the prediction and features is essentially a machine discovered "rule" in the form of a region spanning a named entity with individual linguistic features linking them to a predicted named entity. Hand-made NER rules for Finnish created by a linguist are available in FiNER, which is a rule-based named entity tagger for Finnish. The FiNER system and its technical documentation are available at http://urn.fi/urn:nbn:fi:lb-2018091301. The FiNER rules are based on linguistic features in the named entity or in a sentence surrounding the named entity which can be used to predict the named entity. Analyzing the hand-made rules, one can see which feature configurations correlate with which named entity. The hand-made rules could be transformed into a test set for evaluating the machine discovered results.

3. In this work, the authors introduce an explanation method which can be employed for any sequence-based NLP task. While forming explanations, they treat each feature in a given region as independent from each other. However, as the authors point out, the features can be related to each other in many ways. The analysis of this type of feature structures are left for future work.

As the authors analysis shows and the FiNER rule set displays, linguistic features seldom predict named entity tags by themselves. They usually appear in various configurations or feature structures which may occur both within a word or extend to words in the neighboring context, so some way to extend the analysis to feature structures in a somewhat wider context will need to be explored in future work. Potentially useful feature structures to look for could probably be gleaned from the hand-made FiNER rule set.

4. The language of the paper is in general clear and fluent, but some definite and indefinite articles have been omitted and there are a few minor typos which would be spotted by a native speaker carefully reading the manuscript.

Reviewer #2: In this article, the authors aim to define the concept of region that links the prediction and features that are important for the model. As a case study, the authors apply their method for sequence labelling task, that is named entity recognition. The morphological features are used for the neural named entity recognition model and the important morphological features are analysed for both Turkish and Finnish, being two morphologically rich languages.

The article is well written with all mathematical details and a simple toy example, which clarifies how the features are obtained in the model.

The model features are obtained by applying some perturbations in the regions defined on the data example, which is similar to the idea in Contrastive Estimation (Smith and Eisner). The perturbed examples are fed into a prediction function. This brings the question, what if the size of the feature set is huge? Is it the case for the NER problem? Did you do any experiments by choosing features randomly as you mentioned in your possible solution?

I liked the idea of ranking the morphological features for each named entity type. The results for Finnish and Turkish seem to be coherent which other. The authors could use the obtained important features for the named entity recognition task to analyse if there would be any improvement on the final task. That would be quite interesting, in my opinion. The article seems to be lacking this kind of inference. The findings are interesting but they should be exploited and then the article itself would become complete in that case, in my opinion.

Even the neural architecture could be extended further by injecting the morphological features with their ranks (similar to the attention mechanism), which can be considered as a future work.

6. PLOS authors have the option to publish the peer review history of their article (what does this mean?). If published, this will include your full peer review and any attached files.

Reviewer #1: No

Reviewer #2: No

---

## [Author Response · Author response to Decision Letter 0]

20 Jul 2020

We have attached a document called 'Response to reviewers' which includes our responses to your comments as requested in the e-mail from the editor.

You can find the text below, too. However, due to the capabilities of this form widget, it may not be ideal for reading.

-----

Dear Editor,

 We have attempted to address all the issues that were brought up by the reviewers. We detail the changes that are made in response to each point that is raised by each reviewer. We also explain the additional work done to form these answers regarding each point. We have further performed a complete review of our manuscript to improve the presentation and narrative.

Regards, 

The authors

Point-to-Point responses

The detailed point-to-point responses to the issues identified by the reviewers are provided below.

Reviewer #1:

Dear Reviewer,

We thank you for reviewing our submission and the valuable input and suggestions you provided to improve our manuscript. The following describes our revisions based on your input. We have attempted to respond to all the issues and suggestions raised in your review.

General Comments: In addition to addressing your suggestions, during our revision, we made several editorial changes to improve the focus, scope, clarity and flow of the paper. These revisions are associated with the overall comments of our reviewers and the request for a review by a native speaker. Unless explicitly stated in our point by point responses, the scope of our manuscript has not been altered. In other words, our editorial modifications have been performed only to improve the clarity and accessibility of our work. The file labeled as ‘Revised Manuscript with Track Changes’ is annotated with special markers so that the additions are written in red underlined text while the deletions are written in blue text that is crossed out. Four callout figures in orange were left to indicate the changes that cannot be annotated as others due to technical reasons. One of this is used to indicate the two references added for FiNER, and the other three to indicate new sections.

The following are point to point responses to your review:

Comment 1:

The journal paper states that the joint Finnish NER and MD data set is available at the following repository: https://github.com/onurgu/joint-ner-and-md-tagger/tree/master/dataset.

However, at the time of review, no Finnish data set was available at the location indicated by Güngör et al. In the paper, the authors refer to their own previously published paper using an earlier downloadable version of the named entity annotated data set (https://github.com/mpsilfve/finer-data) developed by a reasearch team in Helsinki, Finland. That data set has a PID and is available through FIN-CLARIN at http://urn.fi/urn:nbn:fi:lb- 2019050201. The data set has recently been published in the Language Resources and Evaluation Journal as "A Finnish news corpus for named entity recognition" (https://link.springer.com/article/10.1007/s10579-019- 09471-7), so a reference to that publication should be added to the list of references as the data set was developed independently of Güngör et al. A post-print of the same journal article is also available at arXiv (https://arxiv.org/abs/1908.04212).

Response 1:

At the time of our submission we were not aware of these publications, which is why we had referred to one of our earlier works that used the datasets. We are pleased to learn that this work has been published and have revised our manuscript to replace the citations to:

33. Ruokolainen T, Kauppinen P, Silfverberg M, Linden K. A Finnish news corpus for named entity recognition. Language Resources and Evaluation. 2019; p. 1–26.

34. University of Helsinki. Finnish News Corpus for Named Entity Recognition; 2019. http://urn.fi/urn :nbn:fi:lb-2019050201.

Comment 2:

The method named EXSEQREG introduces the concept of region that links the prediction and features that are potentially important for the model. A qualitative analysis of the explanations is presented. As the authors point out, there is a lack of established methods for assessing the quality of their predictions.

A region that links the prediction and features is essentially a machine discovered "rule" in the form of a region spanning a named entity with individual linguistic features linking them to a predicted named entity. Hand-made NER rules for Finnish created by a linguist are available in FiNER, which is a rule-based named entity tagger for Finnish. The FiNER system and its technical documentation are available at http://urn.fi/urn:nbn:fi:lb- 2018091301. The FiNER rules are based on linguistic features in the named entity or in a sentence surrounding the named entity which can be used to predict the named entity. Analyzing the hand-made rules, one can see which feature configurations correlate with which named entity. The hand-made rules could be transformed into a test set for evaluating the machine discovered results.

Response 2:

We followed this inspiring recommendation by examining the FiNER rules. We furthermore designed an exper- iment to assess the morphological tags that are highly ranked by our method.

We propose a method that uses FiNER rules to automatically determine the morphological tags relevant to each type of named entity. These tags are compared with the morphological tags deemed significant by our method.

We have revised our manuscript in the following manner to include information about FiNER and this experi- ment:

1. We added the Finnish section under the Using mean normalized importance values for qualitative evalu- ation section to describe the experiment along with FiNER

2. We created a table of FiNER rules with examples (Table 4)

3. We created a table that summarizes the results of the experiment (Table 5)

4. A figure that illustrates how the FiNER rules are used to form a graph of dependencies between rules and

morphological tags (Figure 7)

Comment 3:

In this work, the authors introduce an explanation method which can be employed for any sequence-based NLP task. While forming explanations, they treat each feature in a given region as independent from each other. However, as the authors point out, the features can be related to each other in many ways. The analysis of this type of feature structures are left for future work.

As the authors analysis shows and the FiNER rule set displays, linguistic features seldom predict named entity tags by themselves. They usually appear in various configurations or feature structures which may occur both within a word or extend to words in the neighboring context, so some way to extend the analysis to feature struc- tures in a somewhat wider context will need to be explored in future work. Potentially useful feature structures to look for could probably be gleaned from the hand-made FiNER rule set.

Response 3:

We agree that a further investigation of the dependencies between features is required to be able to better explain the predictions. In this regard, we replaced the part in the Conclusions section with the following paragraph to elaborate more about this issue.

Addition:

Firstly, some morphological tags in a single morphological analysis of a given word are dependent on each other. For instance, the presence of one tag may strongly signal the presence of another tag or the order of appearance in the morpheme sequence may be important. Secondly, this dependence may be observed between features inside and outside the region. For example, named entities are usually related to the features of the words in the left or right contexts, such as the morphological tags and the characters of the surface forms. In future work, we aim to explore such relations by extending our model to permit perturbation across multiple regions of variable sizes. The dependency between features can be explored better if our method allows perturbing one or more features either from the left region, the right region or the region of the named entity itself.

Deletion:

One example of such dependence is when features are ordered sequentially, i.e. a sequence of morpho- logical tags where each one is dependent on the value of the previous ones. The analysis of this type of feature structures are left for future work.

Comment 4:

The language of the paper is in general clear and fluent, but some definite and indefinite articles have been omitted and there are a few minor typos which would be spotted by a native speaker carefully reading the manuscript.

Response 4:

We have done our best to address these issues by carefully reviewing our manuscript to correct typos and grammatical errors.

Reviewer #2:

Dear Reviewer,

We thank you for reviewing our submission and the valuable input and suggestions you provided to improve our manuscript. The following describes our revisions based on your input. We have attempted to respond to all the issues and suggestions raised in your review.

General Comments: In addition to addressing your suggestions, during our revision, we made several editorial changes to improve the focus, scope, clarity and flow of the paper. These revisions are associated with the overall comments of our reviewers and the request for a review by a native speaker. Unless explicitly stated in our point by point responses, the scope of our manuscript has not been altered. In other words, our editorial modifications have been performed only to improve the clarity and accessibility of our work. The file labeled as ‘Revised Manuscript with Track Changes’ is annotated with special markers so that the additions are written in red underlined text while the deletions are written in blue text that is crossed out. Four callout figures in orange were left to indicate the changes that cannot be annotated as others due to technical reasons. One of this is used to indicate the two references added for FiNER, and the other three to indicate new sections.

The following are point to point responses to your review:

Comment 1:

The model features are obtained by applying some perturbations in the regions defined on the data example, which is similar to the idea in Contrastive Estimation (Smith and Eisner). The perturbed examples are fed into a prediction function. This brings the question, what if the size of the feature set is huge? Is it the case for the NER problem? Did you do any experiments by choosing features randomly as you mentioned in your possible solution?

Response 1:

As we note in Section ‘Perturbation’, the number of unique features in the NER task is not very high. For example, in Turkish and Finnish NER tasks, there are 181 and 89 unique features, respectively. Thus, we are able to produce a perturbation for each feature in our work, resulting in a maximum of 181 and 89 perturbed sentences for these languages.

On the other hand, if the task at hand employed a very large number of unique features, as it might be the case

for a sentiment analysis model using ngram frequency features as input, perturbing every feature would require

more time. In such a case, the exact formula of the number of features in a sentence of length N is 2N − 1

when both unigram and bigram features are considered (each feature is composed of either one or two words).

Moreover the choice of the perturbation strategy may result in a large number of features. For example, if the

perturbation is performed by randomly removing K words in the sentence (each feature is a K-word set), the

number of perturbed sentences is equal to C(N,K) = N!/((N−K)!K!) which grows very quickly as N increases. 

In practice, however, this is not a problem as the sentences are not very long. In the NER datasets we used, the average number of words in both Turkish and Finnish sentences was 12. So using the above mentioned equations, when K = 2 a single expansion corresponds to 23 and 66 perturbed sentences, respectively.

We agree that our paper has not properly addressed the issue of huge feature sets. That being said, for pertubra- tion strategies that results in exponentially growing features, a random sampling of possible perturbations (like in [1]) seems reasonable.

To elaborate on the impact of perturbation strategies on the feasibility of the method, the following was added to Section Perturbation to clearly define the case where the number of unique features can be a problem:

Addition:

Unlike the named entity recognition task where M is low, this perturbation strategy might be problematic for other tasks where the number of unique features is very high, e.g. tasks which form features from portions of the input in a combinatorial way so that the number of features grows quickly as the length of the sentence grows. In those cases, a random sample in the vicinity of Xi might be obtained by choosing a feature f from F in a uniformly random fashion and removing it from Xi. This is repeated for a limited number to form the set of perturbed samples with a feasible size.

Deletion:

Unlike named entity recognition task where M ∼ |Fij | in general, this perturbation strategy might be problematic for other tasks where the number of unique features is very high. In those cases, a random sample in the vicinity of Xi is obtained by choosing a feature f from F in a uniformly random fashion and removing it from Xi. This is repeated to form the set of perturbed samples with a feasible size.

To sum up, the answers are:

1. Question: "What if the size of the feature set is huge?": We think that limiting the number of features similar to [1] is appropriate.

2. Question: "Is it the case for the NER problem?": No, the maximum number of perturbations are 181 and 89 for Turkish and Finnish, respectively.

3. Question:"Didyoudoanyexperimentsbychoosingfeaturesrandomlyasyoumentionedinyourpossible solution?": No, we did not. As our perturbation strategy and the number of unique features did not result in an infeasible number of perturbations, we decided to leave it to other works. However, we intend to explore this approach in future work.

1. Ribeiro MT, Singh S, Guestrin C. Why should i trust you?: Explaining the predictions of any classifier. In: Proceedings of the 22nd ACM SIGKDD International Conference on Knowledge Discovery and Data Mining. ACM; 2016. p. 1135–1144.

Comment 2:

I liked the idea of ranking the morphological features for each named entity type. The results for Finnish and Turkish seem to be coherent which other. The authors could use the obtained important features for the named entity recognition task to analyse if there would be any improvement on the final task. That would be quite interesting, in my opinion. The article seems to be lacking this kind of inference. The findings are inter- esting but they should be exploited and then the article itself would become complete in that case, in my opinion.

Even the neural architecture could be extended further by injecting the morphological features with their ranks (similar to the attention mechanism), which can be considered as a future work.

Response 2:

As explained in the paper, we propose a method for determining the most important features of a given model for a given task. We chose a neural NER tagger as a use case and presented the outcome as a ranking among the morphological tags.

It is tempting to exploit this information to be able to train a better NER tagger. However, as the ranking of the morphological tags are obtained by examining a pretrained instance of the NER tagger, these morphological tags are not necessarily the ones that are important universally for the NER task. Thus, if we train a new model while attending to the higher ranked morphological tags from another model, the new model could be biased in the wrong direction.

So, instead of training new models using an attention mechanism, we decided to assess the hypothesis that higher ranked morphological tags can improve the performance for NER by following a corpus-based approach. 

Briefly, we removed a morphological tag from the entire corpus. We then trained a NER tagger with this corpus and recorded the performance. We expect to observe a positive value when the performance corresponding to a ‘related’ tag is subtracted from the performance corresponding to a ‘unrelated’ tag, i.e. removing ‘related tags’ should have lowered the performance more than ‘unrelated tags’. Unfortunately, as you will see in Table 10, the difference between ‘related’ and ‘unrelated’ tags are not always positive as we expected. We suspect the reason for this inconsistency is due to the dependencies between morphological tags.

We added a new section named Effect of the absence of a morphological tag which includes a detailed explana- tion of the experiments and the discussion of the results.

Reviewer #3:

Dear Reviewer,

We thank you for reviewing our submission and the valuable input and suggestions you provided to improve our 

manuscript. The following describes our revisions based on your input. We have attempted to respond to all the issues and suggestions raised in your review.

General Comments: In addition to addressing your suggestions, during our revision, we made several editorial changes to improve the focus, scope, clarity and flow of the paper. These revisions are associated with the overall comments of our reviewers and the request for a review by a native speaker. Unless explicitly stated in our point by point responses, the scope of our manuscript has not been altered. In other words, our editorial modifications have been performed only to improve the clarity and accessibility of our work. The file labeled as ‘Revised Manuscript with Track Changes’ is annotated with special markers so that the additions are written in red underlined text while the deletions are written in blue text that is crossed out. Four callout figures in orange were left to indicate the changes that cannot be annotated as others due to technical reasons. One of this is used to indicate the two references added for FiNER, and the other three to indicate new sections.

The following are point to point responses to your review:

Comment 1:

The paper is disordered. Please, try to rearrange the figures and the tables to be in line with the text.

Response 1:

We understand that this misalignment was caused when we uploaded the figures as separate files to the PLOS submission system. We thought that the system will rename the files according to the upload order.

We corrected this by renaming the files in this revision.

---

## [Decision Letter · Decision Letter 1]

6 Aug 2020

PONE-D-19-26331R1

EXSEQREG: Explaining sequence-based NLP tasks with regions: With a case study using morphological features for named entity recognition

PLOS ONE

Dear Dr. Gungor,

Thank you for submitting your manuscript to PLOS ONE. After careful consideration, we feel that it has merit but does not fully meet PLOS ONE’s publication criteria as it currently stands. Therefore, we invite you to submit a revised version of the manuscript that addresses the points raised during the review process. Please take special attention to the linguistic quality of the manuscript.

We look forward to receiving your revised manuscript.

Kind regards,

Diego Raphael Amancio

Academic Editor

PLOS ONE

Reviewers' comments:

Reviewer's Responses to Questions

**Comments to the Author**

1. If the authors have adequately addressed your comments raised in a previous round of review and you feel that this manuscript is now acceptable for publication, you may indicate that here to bypass the “Comments to the Author” section, enter your conflict of interest statement in the “Confidential to Editor” section, and submit your "Accept" recommendation.

Reviewer #1: (No Response)

Reviewer #3: All comments have been addressed

2. Is the manuscript technically sound, and do the data support the conclusions?

Reviewer #1: Yes

Reviewer #3: Yes

3. Has the statistical analysis been performed appropriately and rigorously? 

Reviewer #1: Yes

Reviewer #3: Yes

4. Have the authors made all data underlying the findings in their manuscript fully available?

Reviewer #1: Yes

Reviewer #3: Yes

5. Is the manuscript presented in an intelligible fashion and written in standard English?

Reviewer #1: No

Reviewer #3: Yes

6. Review Comments to the Author

Reviewer #1: Same research was published ->

Some research was published

in a given region, . ->

in a given region.

impact of each feature to the current prediction ->

impact of each feature on the current prediction

can be referred as an explanation ->

can be referred to as an explanation

Only for the precision metric of ..., the model’s performance is affected ->

Only for the precision metric of ..., is the model’s performance affected

a relatively less number of co-occurring morphological tags ->

a relatively smaller number of co-occurring morphological tags

Reviewer #3: This version of the paper is much better than the previous one. I strongly support its publishing in the journal.

7. PLOS authors have the option to publish the peer review history of their article (what does this mean?). If published, this will include your full peer review and any attached files.

Reviewer #1: No

Reviewer #3: No

---

## [Author Response · Author response to Decision Letter 1]

16 Sep 2020

Dear Editor,

Our last review found our manuscript technically acceptable, however requested a few editorial corrections. We have addressed all of these corrections. Furthermore, we have meticulously reviewed the entire manuscript to improve its grammar and narration. This review was performed by each other as well as an external person. This revision focuses solely on improving the clarity, accessibility and linguistic quality of our manuscript. The scope and technical content of this submission remains the same as the previous version.

We provide a point-to-point response to each reviewer.

Regards, 

The authors

16/09/2020

The detailed point-to-point responses to the issues identified by the reviewers are provided below.

Reviewer #1:

Dear Reviewer,

We thank you for reviewing our submission and the valuable input and suggestions you provided to improve our manuscript. We have attempted to respond to all the issues and suggestions raised in your review.

In addition to addressing your suggestions, we made several editorial changes to improve the clarity and flow of the manuscript. The scope and technical content of our manuscript has not been altered. In other words, our editorial modifications have been performed only to improve the clarity, accessibility and linguistic quality of our work. The file labeled as ‘Revised Manuscript with Track Changes’ is annotated with special markers so that the additions are written in red underlined text while the deletions are written in blue text that is crossed out.

The following are point to point responses to your review:

Comment 1:

Same research was published -> Some research was published 

in a given region, . -> in a given region.

impact of each feature to the current prediction -> impact of each feature on the current prediction 

can be referred as an explanation -> can be referred to as an explanation

Only for the precision metric of ..., the model’s performance is affected -> Only for the precision metric of ..., is the model’s performance affected

a relatively less number of co-occurring morphological tags -> a relatively smaller number of co-occurring morphological tags

Response 1:

Thank you for your corrections. We have corrected all of the errors you have pointed out. Furthermore, we have performed a full review of the paper from an editorial perspective. We have addressed all the grammatical issues identified during this review. Also, we have modified the manuscript to improve the clarity and narrative of our presentation. We believe that these efforts have significantly improved the quality of our presentation.

Reviewer #3:

Dear Reviewer,

We thank you for reviewing our submission and the valuable input and suggestions you provided to improve our manuscript. In this revision we have made several editorial changes to improve the clarity and flow of the manuscript. The scope and technical content of our manuscript has not been altered. In other words, our editorial modifications have been performed only to improve the clarity, accessibility and linguistic quality of our work. The file labeled as ‘Revised Manuscript with Track Changes’ is annotated with special markers so that the additions are written in red underlined text while the deletions are written in blue text that is crossed out.

The following are point to point responses to your review:

Comment 1:

This version of the paper is much better than the previous one. I strongly support its publishing in the journal.

Response 1:

We are very pleased and encouraged by your comment. Thank you for your recommendation.

---

## [Decision Letter · Decision Letter 2]

7 Dec 2020

EXSEQREG: Explaining sequence-based NLP tasks with regions: With a case study using morphological features for named entity recognition

PONE-D-19-26331R2

Dear Dr. Gungor,

We’re pleased to inform you that your manuscript has been judged scientifically suitable for publication and will be formally accepted for publication once it meets all outstanding technical requirements.

Kind regards,

Diego Raphael Amancio

Academic Editor

PLOS ONE

Additional Editor Comments (optional):

Reviewers' comments:

Reviewer's Responses to Questions

**Comments to the Author**

1. If the authors have adequately addressed your comments raised in a previous round of review and you feel that this manuscript is now acceptable for publication, you may indicate that here to bypass the “Comments to the Author” section, enter your conflict of interest statement in the “Confidential to Editor” section, and submit your "Accept" recommendation.

Reviewer #1: All comments have been addressed

Reviewer #4: All comments have been addressed

2. Is the manuscript technically sound, and do the data support the conclusions?

Reviewer #1: Yes

Reviewer #4: Yes

3. Has the statistical analysis been performed appropriately and rigorously? 

Reviewer #1: Yes

Reviewer #4: Yes

4. Have the authors made all data underlying the findings in their manuscript fully available?

Reviewer #1: Yes

Reviewer #4: Yes

5. Is the manuscript presented in an intelligible fashion and written in standard English?

Reviewer #1: Yes

Reviewer #4: Yes

6. Review Comments to the Author

Reviewer #1: (No Response)

Reviewer #4: I believe the paper is good "as is". It is presenting a clear idea and it is exemplifying it using concrete data.

7. PLOS authors have the option to publish the peer review history of their article (what does this mean?). If published, this will include your full peer review and any attached files.

Reviewer #1: No

Reviewer #4: No

---

## [Editor Report · Acceptance letter]

17 Dec 2020

PONE-D-19-26331R2 

EXSEQREG: Explaining sequence-based NLP tasks with regions
With a case study using morphological features for named entity recognition 

Dear Dr. Gungor:

I'm pleased to inform you that your manuscript has been deemed suitable for publication in PLOS ONE. Congratulations! Your manuscript is now with our production department. 

Kind regards, 

on behalf of

Dr. Diego Raphael Amancio 

Academic Editor

PLOS ONE